# Isolation and Characterization of Endophyte *Bacillus velezensis* KOF112 from Grapevine Shoot Xylem as Biological Control Agent for Fungal Diseases

**DOI:** 10.3390/plants10091815

**Published:** 2021-08-31

**Authors:** Kazuhiro Hamaoka, Yoshinao Aoki, Shunji Suzuki

**Affiliations:** Laboratory of Fruit Genetic Engineering, The Institute of Enology and Viticulture, University of Yamanashi, Kofu 400-0005, Japan; g19dia03@yamanashi.ac.jp (K.H.); yaoki@yamanashi.ac.jp (Y.A.)

**Keywords:** *Bacillus velezensis*, downy mildew, gray mold, ripe rot, PR protein, zoosporangia

## Abstract

As the use of chemical fungicides has raised environmental concerns, biological control agents have attracted interest as an alternative to chemical fungicides for plant-disease control. In this study, we attempted to explore biological control agents for three fungal phytopathogens causing downy mildew, gray mold, and ripe rot in grapevines, which are derived from shoot xylem of grapevines. KOF112, which was isolated from the Japanese indigenous wine grape *Vitis* sp. cv. Koshu, inhibited mycelial growth of *Botrytis cinerea*, *Colletotrichum gloeosporioides*, and *Phytophthora infestans*. The KOF112-inhibited mycelial tips were swollen or ruptured, suggesting that KOF112 produces antifungal substances. Analysis of the 16S rDNA sequence revealed that KOF112 is a strain of *Bacillus velezensis*. Comparative genome analysis indicated significant differences in the synthesis of non-ribosomal synthesized antimicrobial peptides and polyketides between KOF112 and the antagonistic *B. velezensis* FZB42. KOF112 showed biocontrol activities against gray mold caused by *B. cinerea*, anthracnose by *C. gloeosporioides*, and downy mildew by *Plasmopara viticola*. In the KOF112–*P. viticola* interaction, KOF112 inhibited zoospore release from *P. viticola* zoosporangia but not zoospore germination. In addition, KOF112 drastically upregulated the expression of genes encoding class IV chitinase and β-1,3-glucanase in grape leaves, suggesting that KOF112 also works as a biotic elicitor in grapevine. Because it is considered that endophytic KOF112 can colonize well in and/or on grapevine, KOF112 may contribute to pest-management strategies in viticulture and potentially reduce the frequency of chemical fungicide application.

## 1. Introduction

The production of Koshu wine in Japan started in 1874 in Yamanashi Prefecture. *Vitis* sp. cv. Koshu, a hybrid of *Vitis vinifera* L. and *V. davidii* Foex, is an indigenous wine grape in Japan [1]. Koshu was introduced from Europe to Japan through the Silk Road, and crossed with the Chinese wild species *V. davidii* en route to Japan. Koshu was recognized as a wine grape cultivar in 2010 by the International Organization for Vine and Wine, and was registered in *Vitis* International Variety Catalogue by Julius Kühn-Institut-Bundesforschungsinstitut für Kulturpflanzen (https://www.vivc.de/, accessed on 28 April 2021). At present, Koshu is one of the most widely cultivated wine grapes in Japan and one of the most important cultivars for white-wine making in Japan [2]. Whole genome analysis demonstrated that Koshu is susceptible to phytopathogenic attack as a result of deletions in genes associated with pathogen response, such as hypersensitive response [3]. For example, Koshu is more susceptible to downy mildew, which is caused by *Plasmopara viticola* (Berk and M.A. Curtis; Berl and De Toni), than other *V. vinifera* cultivars [4]. In addition, the humid climate of Japan, caused by the prolonged concentration of extremely moist airstreams over Japan as a result of global warming [5], has contributed to damaging Koshu grapevines by phytopathogenic fungal diseases. The high nighttime temperature resulting from the acceleration of global warming has promoted downy mildew infection in grapevines [6]. Thus, fungal disease control has become a subject of serious concern in viticulture for Koshu wine making.

A simple strategy against fungal disease is the use of chemical fungicides. However, there are two main problems with regard to the application of chemical fungicides. One is environmental pollution, and the other is the emergence of fungal phytopathogen populations resistant to the chemical fungicides. In particular, the latter has plagued vine growers in Japan, making it extremely difficult to control fungal diseases in grapevines. *P. viticola* is a high-risk phytopathogen that easily acquires chemical fungicide resistance [7]. Resistant genes, conferring resistance to quinone outside inhibitor and carboxylic acid amide, were detected in *P. viticola* populations in Japanese vineyards in 2010 [8] and in 2015 [9], respectively. Fungicide-resistant *Colletotrichum gloeosporioides* (Penzig) Penzig and Saccardo, which causes grape ripe rot, and *Botrytis cinerea* Pers. ex Fr., which causes grape gray mold, have already emerged in Japanese vineyards [10,11]. It is for these reasons that alternatives to chemical fungicides are attracting the interest of scientific communities.

One of the alternative disease control strategies to chemical fungicides is biological control using biofungicides. Biological control agents in biofungicides are isolated from nature, and are largely microorganisms [12]. A vast number of microorganisms isolated from nature have been identified as candidates for biological control agents in biofungicides [13,14]. Some microorganisms have been developed and launched as biofungicides in viticulture. For example, *Bacillus subtilis* QST-713 (product name, Serenade^®^) is available on the market, and is used to control gray mold in viticulture [15]. The introduction of biofungicide application in viticulture is expected to reduce the frequency of chemical fungicide application and to alleviate increasing environmental concerns on chemical fungicides and the emergence of resistance to the chemical fungicides in fungal phytopathogens.

Cyclic lipopeptides produced by biological control agents have received considerable attention as one of the tools for disease control in plants, because some cyclic lipopeptides also function as elicitors in plants as well as antimicrobial metabolites [16,17,18]. For example, fengycin and surfactin secreted by *B. subtilis* GLB191 contribute to protection against grape downy mildew by directly inhibiting *P. viticola* and inducing plant defense response [17]. Iturin A induces defense response in plants depending on its structure [19]. The cyclization of the seven amino acids and the β-hydroxy fatty acid chain of iturin A are required for the induction of plant defense response. Although evidence of how cyclic lipopeptides trigger plant defense response is lacking, microorganisms showing bifunctional activity against phytopathogens may be an innovative biological control agent in viticulture.

The objective of this study was to clarify the possibility of using endophytic bacteria as biological control agents in biofungicides used in viticulture. The colonization efficiency of biological control agents in biofungicides on/in plant tissues affects their antagonistic activities toward fungal phytopathogens [20]. In this study, we explored grapevine endophytic bacteria possessing in vitro antagonistic activities toward three fungal phytopathogens, *B. cinerea*, *C. gloeosporioides*, and *Phytophthora infestans* (a substitute of *P. viticola*), and isolated endophytic *Bacillus velezensis* KOF112 from the shoot xylem of Koshu grapevine. KOF112 showed in vivo biocontrol activities against gray mold caused by *B. cinerea*, anthracnose by *C. gloeosporioides*, and downy mildew by *P. viticola*. In addition, foliar application of KOF112 induced plant defense response in grapevine.

## 2. Results

### 2.1. Antagonistic Activity of KOF112 toward Phytopathogenic Fungi

Two hundred and forty-seven colonies were collected from Koshu shoot xylem and subjected to the in vitro bioassay using *B. cinerea*, *C. gloeosporioides*, and *P. infestans*. As *P. viticola* is an obligate biotrophic oomycete, *P. infestans* was used as the oomycete phytopathogen instead of *P. viticola*. As a result, one colony that exhibited strong antagonistic activity toward the three phytopathogenic fungi was successfully isolated and named KOF112. Large inhibition zones were formed between KOF112 and each phytopathogenic fungus (Figure 1A). The mycelial tips of growth-inhibited *B. cinerea*, *C. gloeosporioides*, and *P. infestans* by KOF112 were swollen or ruptured (Figure 1B). *Agrobacterium* sp. isolate CHB3, selected as the control isolate with no antifungal activity, exhibited no suppressive effect on the mycelial growth of all the fungi tested.

These results suggest that KOF112 inhibits the mycelial growth of all the fungi tested, and that KOF112 produces antifungal substances.

### 2.2. KOF112 Is a Strain of Bacillus velezensis

The 16S rDNA nucleotide sequence of KOF112 had high homologies to the corresponding sequences of *B. velezensis* BIM B-1312D (100%), *B. velezensis* FJAT-52631 (100%), and *B. velezensis* CR-502 (99.9%). Phylogenetic analysis of the nucleotide sequences of KOF112 and *Bacillus* isolates showed that KOF112 formed a cluster with *B. velezensis* FZB42, which has a capacity to produce antimicrobial secondary metabolites (Figure 2) [21].

The draft genome of KOF112 includes a circular genome (3,916,789 bp) and a plasmid (13,003 bp). The annotation predicted 3746 coding sequences, 27 rRNA genes, and 86 tRNA genes. Comparison of KOF112 genome sequence with the genome sequences of antagonistic *Bacillus* isolates, *B. velezensis* FZB42 and *B. amyloliquefaciens* DSM7, which formed a cluster by 16S rDNA comparison (Figure 2), revealed significant differences among the genome sequences (Figure 3). For example, the gene clusters responsible for the non-ribosomal synthesized antimicrobial peptides and polyketides in KOF112 genome were different from those in FZB42 and DSM7 genomes. KOF112 genome has gene clusters for the biosynthesis of surfactin, bacillibactin, bacilysin, and bacillaene, but not gene clusters for the biosynthesis of iturin, bacillomycin, fengycin, difficidin, and macrolactin.

Surfactin synthase subunit 1 (*srfAA*), surfactin synthase subunit 2 (*srfAB*), surfactin synthase subunit 3 (*srfAC*), and surfactin synthase thioesterase subunit (*srfAD*) were present in KOF112 genome (Figure 3). Surfactin biosynthetic gene cluster was selected as a non-ribosomal peptide synthase cluster and compared among KOF112, *B. velezensis* FZB42, and *B. amyloliquefaciens* DSM7 (Figure 4). The organization of *srfAA*, *srfAB*, *srfAC*, and *srfAD* showed a perfect match among the isolates (Figure 4A). Phylogenetic analysis of each gene indicated that KOF112 surfactin biosynthetic genes formed a cluster with *B. velezensis* FZB42 surfactin biosynthetic genes (Figure 4B).

Taken together, the results suggest that KOF112 is a strain of *B. velezensis*.

### 2.3. Biocontrol Activity of KOF112 against Downy Mildew in Grapevine

Because KOF112 inoculum contained 10% soybean casein digest (SCD) medium, 10% SCD medium was used as a control in the assay for biocontrol activity of KOF112. A large number of *P. viticola* white symptoms were observed on the untreated disks and the disks treated with SCD, whereas growth of symptoms on the disks treated with KOF112 was apparently inhibited (Figure 5A). In particular, 1 × 10^8^ cfu/mL KOF112 completely controlled *P. viticola* symptoms. Disease severity was significantly reduced by KOF112 compared with control or SCD treatment (Figure 5B). The inhibitory effect of KOF112 on *P. viticola* infection was dose-dependent. No symptoms were observed on the disks treated with 1 × 10^8^ cfu/mL KOF112, whereas a large number of white symptoms were visible on the disks treated with 1 × 10^5^ cfu/mL KOF112. *Agrobacterium* sp. isolate CHB3 exhibited no inhibitory effect on downy mildew.

### 2.4. Biocontrol Activity of KOF112 against Gray Mold in Cucumber

Cucumber leaves were used as the alternative host of *B. cinerea* because it was difficult to evaluate *B. cinerea* symptoms on grapevine leaves. The cucumber-*B. cinerea* pathosystem was used. Large and severe symptoms caused by *B. cinerea* were clearly observed on control or SCD-treated leaves (Figure 6). KOF112 (1 × 10^8^ cfu/mL) significantly reduced the disease severity caused by *B. cinerea* compared with control and SCD treatment (Figure 6). *Agrobacterium* sp. isolate CHB3 exhibited no inhibitory effect on gray mold.

### 2.5. Biocontrol Activity of KOF112 against Anthracnose in Strawberry

Strawberry leaves were used as the alternative host of *C. gloeosporioides* because it was difficult to evaluate *C. gloeosporioides* symptoms on grapevine leaves. The strawberry-*C. gloeosporioides* pathosystem was used. Large symptoms caused by *C. gloeosporioides* were clearly observed on control or SCD-treated leaves (Figure 7). KOF112 (1 × 10^8^ cfu/mL) significantly reduced the disease severity caused by *C. gloeosporioides* compared with control and SCD treatment (Figure 7). *Agrobacterium* sp. isolate CHB3 exhibited no inhibitory effect on anthracnose.

### 2.6. Inhibition of Zoospore Release from P. viticola Zoosporangia by KOF112

As KOF112 inhibited the mycelial growth of *B. cinerea*, *C. gloeosporioides*, and *P. infestans* (Figure 1), mycelial disks of *B. cinerea* and *C. gloeosporioides* were used as inocula in the bioassays for biocontrol activity of KOF112 against gray mold and anthracnose, respectively (Figure 6 and Figure 7). On the other hand, we used *P. viticola* zoosporangia as inoculum in the bioassay for biocontrol activity of KOF112 against downy mildew (Figure 5). Zoosporangia release many zoospores after inoculation, and the zoospores penetrate leaves through stomata [22]. Microscopic observation was performed to evaluate the effect of KOF112 on the early infection behaviors of *P. viticola* zoosporangia and zoospores. Empty zoosporangia were observed in the case of zoosporangia not treated or treated with SCD within 24 h after the treatment, whereas a large number of zoosporangia treated with KOF112 still had zoospores inside them after 24 h (Figure 8A). Interestingly, KOF112 seemed to surround zoosporangia that did not release zoospores (Figure 8A). Approximately 50% of zoosporangia not treated or treated with SCD released zoospores 24 h after the treatment, whereas approximately 24% of zoosporangia treated with KOF112 released zoospores after the same treatment period (Figure 8B). On the other hand, approximately 45% of zoospores germinated in spore suspension not treated or treated with 10% SCD after incubation for 20 h, whereas approximately 50% of zoospores germinated in the presence of KOF112 (Figure 8C). *Agrobacterium* sp. isolate CHB3 exhibited no effect on both zoospore release from zoosporangia and zoospore germination.

These results suggest that KOF112 inhibits zoospore release from *P. viticola* zoosporangia but not zoospore germination.

### 2.7. KOF112 Induces Plant Defense Response in Grapevine

The possibility that KOF112 induced plant defense response in grapevine was evaluated. We selected genes encoding class IV chitinase and β-1,3-glucanase as indicators of plant defense response, because chitinase and β-1,3-glucanase gene expression is induced through jasmonic acid (JA) and salicylic acid (SA)-dependent defense pathways, respectively. KOF112 drastically upregulated the expression of a gene-encoding class IV chitinase in grape leaf disks 48 h after KOF112 treatment, compared with untreated control and SCD treatment (Figure 9). The transcripts of β-1,3-glucanase gene were also increased 24 and 48 h after KOF112 treatment compared with untreated control and SCD treatment (Figure 9). *Agrobacterium* sp. isolate CHB3 did not upregulate the transcription of both genes.

These results suggest that KOF112 also works as a biotic elicitor and induces plant defense response in grapevine through both JA- and SA-dependent defense pathways.

## 3. Discussion

Grapes are one of the most important fruits cultivated worldwide. However, their high susceptibility to pre- and post-harvest pathogens has resulted in significant economic losses. The main diseases in viticulture are gray mold [23], ripe rot [24], and downy mildew [25]. As a biological control agent, KOF112 is able to suppress those three diseases. Although the disease-suppressing effect of biological control agents is weaker than that of chemical fungicides, the application of KOF112 having antagonistic activities toward a wide range of phytopathogenic fungi, including *Ascomycetes* and *Oomycetes*, may contribute to reducing the frequency of chemical fungicide application in viticulture, as well as to inhibiting the development of chemical fungicide resistance. The supernatants of biological control agents could be used as a new biostimulant in sustainable agriculture [26]. The supernatant of *B. subtilis* GLB191, which contains surfactin and fengycin, exerted a direct antifungal effect and induced plant defense response, thereby contributing to protection against grape downy mildew [17]. Future laboratory and field trials are necessary to evaluate whether the supernatant of KOF112 can be used as a biostimulant against gray mold, ripe rot, and downy mildew, and to identify the optimum conditions, including the dose of KOF112 and the timing of KOF112 application, to ensure that the combination treatment of KOF112 and chemical fungicides works effectively in viticulture.

Most commercially available biological control agents are *Bacillus*, which produces a large number of antibiotics [27,28]. *B. velezensis* FZB42 (former name *B. amyloliquefaciens* subsp. *plantarum*) [29] is used commercially as a biological control agent in agriculture [30] and produces such metabolites as peptides and polyketides, having antifungal, antibacterial, and nematocidal activities [31]. However, doubtful evidence related to the direct antifungal activity of FZB42 metabolites against competing plant pathogens has been presented. The concentrations of the metabolites produced by FZB42 in plants [32] were relatively low and/or undetectable, with the exception of surfactin. On the other hand, FZB42 compensated changes in microbial community structure caused by pathogens and helped plant-associated *Bacilli* contribute to plant protection [33]. Consequently, the sub-lethal concentrations of cyclic lipopeptides produced by *Bacilli* triggered plant defense response systemically.

Cyclic lipopeptides were found to enhance plant defense response to phytopathogenic fungi [34,35]. Surfactin and fengycin protected grape against downy mildew by exerting a direct antifungal effect and inducing plant defense response [17]. Cyclic lipopeptides activated plant defense response through distinct defense pathways [36]. Mycosubtilin (iturin family) activated both JA- and SA-dependent defense pathways, whereas surfactin mainly induced an SA-dependent plant defense response [36]. In the present study, gene expression analysis suggested that KOF112 induced plant defense response in grapevine through both JA- and SA-dependent defense pathways. Thus, enhancing plant defense response by antifungal metabolites produced by biological control agents is one of the mechanisms for protecting plants. Future studies employing liquid chromatography-tandem mass spectrometry analysis would identify the cyclic lipopeptides produced by KOF112 and reveal which cyclic lipopeptides contribute to the protective effect of KOF112 against fungal diseases.

In this study, we found that KOF112 also induced plant defense response in grapevine. Chitinase inhibited infection by *B. cinerea* [37] and *C. gloeosporioides* [38], whereas β-1,3-glucanase exerted direct antimicrobial activity against *B. cinerea* [39], *C. gloeosporioides* [40], and *P. viticola* [41]. Genome sequencing was attempted to clarify the biocontrol activity of the biological control agents [42]. Comparative genome analysis of KOF112 and FZB42 demonstrated that KOF112 might produce fewer antimicrobial peptides and polyketides than FZB42. The direct inhibition of phytopathogenic fungal mycelial growth in vitro is an indirect indication that KOF112 produces some antibiotics. Although the inhibition of zoospore release from *P. viticola* zoosporangia by KOF112 may be an interesting mechanism underlying the biocontrol activity against grape downy mildew, the probable antibiotic production by KOF112 may confer enhanced plant defense response in the same way as FZB42. Further studies employing the genomic analysis of KOF112 for exploring genes encoding bioactive substances, as well as the qualitative and quantitative analyses of antibiotics produced by KOF112, would reveal the main mechanism underlying plant disease control by KOF112. Because we were unable to analyze the biocontrol activity of KOF112 by field trials in vineyards, we could not verify whether our laboratory experiments using leaf disks support our hypothesis of increased plant resistance. We need to conduct field trials in vineyards to verify our hypothesis that KOF112 also works as a biotic elicitor in pest management strategies.

Grapevines are a rich source of potential biological control agents for fungal and oomycete pathogens. Bacterial isolates collected from endophytic and epiphytic communities living in grapevine leaves inhibited *B. cinerea* and *P. infestans* mycelial growth [43]. The colonization of biological control agents *in planta* is one of the driving forces for protecting plants against diseases. The suppressive activities of biological control agents against phytopathogenic fungi are influenced by the capacity of those agents to colonize in planta [20,44]. Endophytes are considered potential biological control agents because of their colonization ability. In plant root endospheres, high motility as well as enhanced plant cell-wall degradation and reactive oxygen-species-scavenging abilities seem to be important traits for successful endophytic colonization [45]. Because endophytic bacteria play a role in the resistance to biotic and abiotic stresses as well as plant growth and development, manipulating endophytic bacteria would help us develop novel and innovative techniques for improving agricultural production. In fact, plants have unique endophytes, some of which confer resistance to the plants themselves [46]. In this study, we focused on endophytic bacteria and isolated antifungal endophyte KOF112 from grapevine shoot xylem. It is reasonable to assume that endophytes show greater affinity for plants than soil microorganisms. Although KOF112 can induce plant defense response in grapevine, we cannot present concrete results of KOF112 colonization in grapevine at the moment. Further investigation by scanning electron microscopy would reveal whether KOF112 colonizes foliar-sprayed leaves. If KOF112 can colonize well in and/or on grapevine, KOF112 would have an edge over commercial biological fungicides that are generally considered to have a short life span on grapevine. Future studies involving field trials of KOF112 application to bunches and leaves and/or KOF112 injection into grapevine shoot xylems would reveal whether KOF112 colonization in grapevine results in the optimal biocontrol activity of endophytic KOF112 against fungal diseases.

## 4. Materials and Methods

### 4.1. Plant Materials

Grapevines (*Vitis* sp. cv. Koshu, *V. vinifera* cvs. Pinot Noir, Chardonnay, and Cabernet Sauvignon) were cultivated in the experimental vineyard of The Institute of Enology and Viticulture, University of Yamanashi, Yamanashi, Japan (latitude 35.680528, longitude 138.569268, elevation 250 m). The grapevines were trained in double cordon style and were approximately 30 years old.

### 4.2. Isolation of Endophytic Bacteria from Grapevine Shoot Xylem

Grapevine shoots were collected on 9 December 2019. The surface of the shoots was sterilized with 0.1% (*v*/*v*) sodium hypochlorite solution for 3 min at room temperature. Bark and epidermal tissue were peeled off from the shoots using a knife sterilized with 70% ethanol. Xylem was shaved using a grater sterilized with 70% ethanol. One gram of shaved xylem was placed in a sterilized 100 mL flask containing 40 mL of phosphate buffer solution (pH 7.4). After shaking at 130 rpm for 3 h at 25 °C, the phosphate buffer solution with shaved xylem was filtered through sterilized cotton gauze. One hundred microliters of filtrate was plated on SCD, Luria–Bertani broth, and potato dextrose broth agar (PDA) plates, and the plates were incubated at 25 °C for 3 d. Bacterial isolates that grew on the plates were used in the in vitro bioassay.

### 4.3. In Vitro Bioassay

To screen bacterial isolates for their antagonistic activity toward three fungal phytopathogens, *B. cinerea*, *C. gloeosporioides*, and *P. infestans* (an oomycete pathogen of late blight), the dual culture technique was performed as described previously (Figure 1) [47]. *B. cinerea* and *C. gloeosporioides* laboratory strains isolated from the experimental vineyard of The Institute of Enology and Viticulture were used [11,48]. *P. infestans* isolate (NBRC 9173) was obtained from the NITE Biological Resource Center. Briefly, bacterial isolates were streaked on the edge of SCD plates. An agar plug (6.5 mm diameter) cut from colonies of the fungal phytopathogens was placed at the center of the bacteria-growing SCD plates, and the plates were incubated at 25 °C for 5 d. Bacterial isolates that suppressed the mycelial growth of all the fungal phytopathogens by forming a growth inhibition zone were selected as potent antagonists. From the assay, we obtained KOF112 as the best candidate strain. *Agrobacterium* sp. isolate CHB3 was selected as the control isolate, with no antifungal activity to evaluate the biocontrol activity of KOF112 against the three fungal phytopathogens.

### 4.4. Identification of KOF112 by 16S rDNA Sequence Analysis

Genomic DNA was extracted from the one-day culture of KOF112 in SCD medium using a DNeasy PowerSoil Kit (Qiagen, Hilden, Germany) in accordance with the manufacturer’s instructions. PCR conditions for amplifying partial 16S rDNA were as follows: after incubation at 94 °C for 5 min, PCR amplification was performed for 30 cycles at 94 °C for 30 s, 55 °C for 30 s, and 72 °C for 1 min, with a final extension step at 72 °C for 7 min. The nucleotide sequences of the primer set for amplifying partial 16S rDNA from bacterial genome were as follows: 795F (5′-GGATTAGATACCCTGGTA-3′) and 1492R (5′-GGYTACCTTGTTACGACTT-3′). The nucleotide sequences of the amplicons were analyzed using the dye terminator method and subjected to the Basic Local Alignment Search Tool (BLAST, NCBI). Phylogenetic analysis was performed using the partial 16S rDNA of KOF112 and *Bacillus* isolates deposited in the NCBI database. The nucleotide sequences were subjected to the NJ method using Molecular Evolutionary Genetics Analysis software, MEGA10 (www.megasoftware.net, accessed on 7 April 2021).

### 4.5. Genome Sequencing, Assembly, and Annotation

DNA extraction from KOF112 was performed using a DNeasy PowerSoil Kit (Qiagen). Genome sequencing, assembly, and annotation were performed as described previously [49]. Briefly, short-read sequencing using a DNBSEQ-G400 sequencer (MGI Tech, Shenzhen, China) and long-read sequencing using the GridION nanopore sequencing platform (Oxford Nanopore Technologies, Oxford, UK) were performed in accordance with the manufacturers’ instructions. The high-quality short-read and long-read sequences were assembled using Unicycler (ver. 0.4.7) under default conditions. The annotation was performed by Prokka (ver. 1.13). The draft genome sequence of KOF112 was deposited in DDBJ/ENA/GenBank under accession no. AP024603 (genome) and AP024604 (plasmid).

### 4.6. Comparative Genome Analysis

Genome sequences of *B. velezensis* FZB42 (former name *B. amyloliquefaciens* subsp. plantarum, accession no. CP000560) and *B. amyloliquefaciens* DSM7 (accession no. FN597644) were used as representative antagonistic *Bacillus* species for comparative genome analysis. Comparative genome analysis was performed using the CGView Server [50].

### 4.7. Biocontrol Activity of KOF112 against Downy Mildew

An in vivo bioassay using grape leaf disks was performed as described previously, with minor modifications [51]. Briefly, Koshu leaves were collected from potted seedlings. Five leaf disks each having a diameter of 13 mm were cut out from the leaves using a cork polisher and placed upside down on moistened filter paper in square Petri dishes (140 mm × 100 mm). KOF112 was incubated in SCD medium overnight and adjusted to 1 × 10^8^, 1 × 10^7^, 1 × 10^6^, and 1 × 10^5^ cfu/mL with SCD medium and sterile water (all KOF112 solutions contained 10% SCD medium). Ten microliters of KOF112 solution was dropped onto four locations on the abaxial surface of the leaf disk. Ten percent SCD medium and *Agrobacterium* sp. isolate CHB3 (1 × 10^8^ cfu/mL, containing 10% SCD) were used as control. The leaf disks were dried at room temperature in a flow cabinet for 3 h.

Inoculation of *P. viticola* zoosporangia was performed as described previously with minor modifications [6]. Briefly, field-isolated *P. viticola* was maintained on leaves of potted Koshu seedlings in a growth chamber under light irradiation (11.8 Wm^−2^/16 h/d) at 21 °C. Fresh zoosporangia of *P. viticola* were washed off with sterile water from the symptoms on the Koshu leaves. Ten microliters of the zoosporangium suspension (1 × 10^4^ zoosporangia/mL) was dropped onto the same locations as the pretreated KOF112 solutions on the leaf disks. The Petri dishes containing the leaf disks were placed in a plastic box containing moistened paper towel. The box was incubated in the dark for 24 h and then in an incubator (21 °C, 11.8 Wm^−2^/16 h/d). Downy mildew symptoms on each disk were assessed 7 days post *P. viticola* inoculation. Disease severity was scored by evaluating the symptom on each disk as described previously [51]: 0, no symptoms; 1, white symptom occupies up to 1/6 of the disk; 2, white symptom occupies up to 1/3 of the disk; 3, white symptom occupies up to half of the disk; 4, white symptom occupies up to two-thirds of the disk; 5, white symptom occupies more than two-thirds of the disk. Two independent experiments were performed with five leaf disks.

### 4.8. Biocontrol Activity of KOF112 against Gray Mold

Three true leaves (approximately 4 cm in size) were detached from cucumber (*Cucumis sativus* cv. Sharp-1) seedlings and sprayed with 500 μL of KOF112 solution (1 × 10^8^ cfu/mL, containing 10% SCD) with a hand sprayer. As the control experiment, sterile water, 10% SCD medium, or *Agrobacterium* sp. isolate CHB3 (1 × 10^8^ cfu/mL, containing 10% SCD) was sprayed. After air drying the cotyledons, the center of each cotyledon was punctured with a sterile needle. Mycelial disks (6.5 mm diameter) of *B. cinerea* laboratory strain were excised from the leading edge of the colonies on PDA plates, and a disk was placed on the wound of each cotyledon. After incubation in a moisture chamber at 25 °C for 6 d in an incubator (11.8 Wm^−2^/16 h/d), disease severity was scored by evaluating the symptom of each cotyledon as described previously [11]: 0, no symptoms; 0.5, rot only under inoculum; 1, rot two times larger than the disk; 2, rot three times larger than the disk; 3, rot four times larger than the disk; 4, rot more than five times larger than the disk. Two independent experiments were performed with three leaves.

### 4.9. Biocontrol Activity of KOF112 against Anthracnose

Strawberry leaves (*Fragaria ananassa* cv. Hokowase) were detached from seedlings cultivated in a growth chamber. Three detached leaves were sprayed with 500 μL of KOF112 solution (1 × 10^8^ cfu/mL) with a hand sprayer. As the control experiment, sterile water, 10% SCD medium, or *Agrobacterium* sp. isolate CHB3 (1 × 10^8^ cfu/mL, containing 10% SCD) was sprayed. After air drying the leaves, the center of each leaf was punctured with a sterile needle. Mycelial disks (6.5 mm diameter) of *C. gloeosporioides* laboratory strain were excised from the leading edge of the colonies on PDA plates, and a disk was placed on the wound of each leaf. After incubation in a moisture chamber at 25 °C for 8 d in the dark, disease severity was scored by evaluating the symptom of each leaf as described previously [48]: 0, no symptoms; 1, rot only under the disk; 2, rot two times larger than the disk; 3, rot three times larger than the disk; 4, rot four times larger than the disk; 5, rot more than five times larger than the disk; 6, rot more than six times larger than the disk; 7, rot more than seven times larger than the disk. Four independent experiments were performed with three leaves.

### 4.10. Light Microscope Observation of P. viticola Zoospore Release from Zoosporangia and Zoospore Germination

A zoosporangium suspension of *P. viticola* (1 × 10^4^ zoosporangia/mL) was prepared as mentioned above. Nine hundred microliters of the zoosporangium suspension was incubated with 100 μL of KOF112 solution (1 × 10^8^ cfu/mL, containing 10% SCD), 10% SCD medium, or *Agrobacterium* sp. isolate CHB3 (1 × 10^8^ cfu/mL, containing 10% SCD) in a microtube at 22 °C for 1 h under light irradiation (11.8 Wm^−2^), and then for 23 h in the dark. Zoospore release from zoosporangia was observed under a light microscope (Olympus BX51, Tokyo, Japan). Zoospore release rate was calculated using the following formula:zoospore release (%) = number of empty zoosporangia/number of total zoosporangia × 100

Zoospore germination was counted as described previously, with minor modifications [6]. Briefly, a zoosporangium suspension of *P. viticola* (1 × 10^4^ zoosporangia/mL) was prepared as mentioned above. Five hundred microliters of the zoosporangium suspension was incubated in a microtube at 30 °C for 4 h under light irradiation (11.8 Wm^−2^). One hundred microliters of KOF112 solution (1 × 10^8^ cfu/mL, containing 10% SCD), 10% SCD, or *Agrobacterium* sp. isolate CHB3 (1 × 10^8^ cfu/mL, containing 10% SCD) was added into the microtube, and then incubation was carried out at 30 °C for 12 h under light irradiation (11.8 Wm^−2^), followed by incubation at 30 °C for 8 h in the dark. The suspension was stained with 0.05% aniline blue in 0.0067M K_2_HPO_4_ (pH 9–9.5) at room temperature for 20 min. Zoospore germination was observed using a fluorescence microscope (Olympus BX51). Zoospore germination rate was calculated using the following formula:germination (%) = number of germinated zoospores/number of total zoospores × 100

### 4.11. Real-Time RT-PCR

To determine whether KOF112 induces plant defense response in grapevines, transcriptional alteration of genes encoding two PR proteins, class IV chitinase and β-1,3-glucanase, was evaluated in KOF112-treated grapevine leaves. Three leaf disks each having a diameter of 13 mm were cut out from Koshu leaves using a cork polisher and placed upside down on moistened filter paper in square Petri dishes (140 mm × 100 mm). Ten microliters of KOF112 solution (1 × 10^8^ cfu/mL, containing 10% SCD) was dropped onto four locations on the abaxial surface of the leaf disk. Ten percent SCD medium or *Agrobacterium* sp. isolate CHB3 (1 × 10^8^ cfu/mL, containing 10% SCD) was used as control. After incubation at 22 °C for 24 h and 48 h in an incubator (11.8 Wm^−2^/16 h/d), the disks were homogenized in a mortar containing liquid nitrogen using a pestle. Total RNA isolation from the pulverized samples was performed using NucleoSpin RNA Plant (Takara, Shiga, Japan), and purification was carried out using Fruit-mate for RNA Purification (Takara). First-strand cDNA was synthesized from the total RNA using PrimeScript RT Master Mix (Perfect Real Time) (Takara). Real-time RT-PCR was performed using an SYBR Premix Ex Taq II (Perfect Real Time) (Takara) with a Thermal Cycler Dice Real Time System (Takara). PCR conditions were as follows: incubation at 95 °C for 30 s, followed by 40 cycles at 95 °C for 5 s and at 60 °C for 45 s. The primers used for amplification were as follows: ubiquitin primers (5′-GTGGTATTATTGAGCCATCCTT-3′ and 5′-AACCTCCAATCCAGTCATCTAC-3′, GenBank accession no. BN000705); class IV chitinase primers (5′-CAATCGGGTCCTTGTGATTC-3′ and 5′-CAAGGCACTGAGAAACGCT-3′, GenBank accession no. U97522), and β-1,3-glucanase primers (5′-GAATCTGTTCGATGCCATGC-3′ and 5′-GCATTATCAACCGTAGTCCC-3′, GenBank accession no. DQ267748). Ubiquitin primers were used as the reference gene to normalize each gene expression because ubiquitin gene expression was stable in the grapevine leaves [52]. Using the standard curve method of Thermal Cycler Dice Real Time System Single Software ver. 3.00 (Takara), gene expression levels were determined as the number of amplification cycles needed to reach a fixed threshold and are expressed as relative values to ubiquitin.

### 4.12. Statistical Analysis

Data are presented as means ± standard deviations of biological replicates. Statistical analysis was performed by the Student’s *t*-test or Dunnett’s multiple comparison test using Excel Statistics software 2012 (Social Survey Research Information, Tokyo, Japan).

## Figures and Tables

**Figure 1 plants-10-01815-f001:**
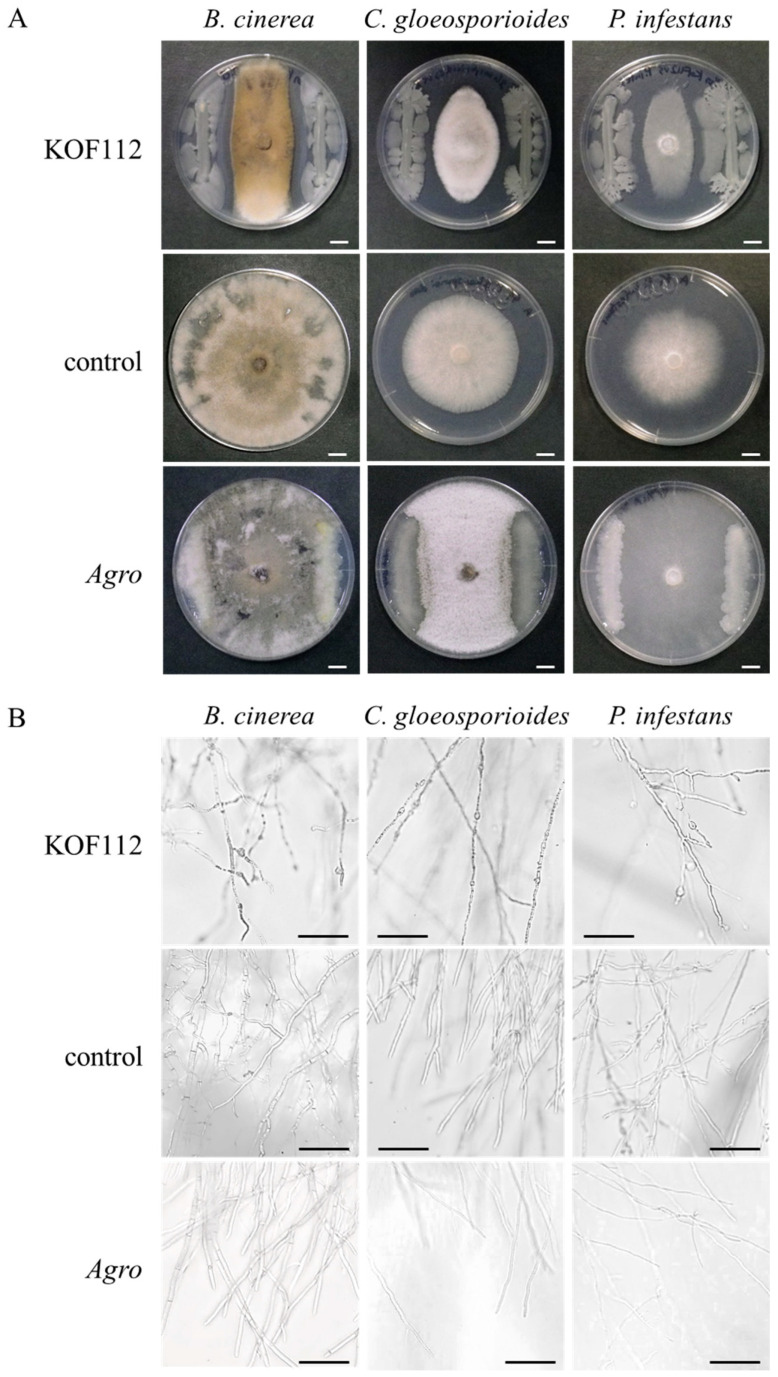
Antagonistic activity of KOF112 toward mycelial growth of phytopathogenic fungi. (**A**) KOF112 was streaked on the edge of SCD plates, and agar plugs cut from colonies of *B. cinerea*, *C. gloeosporioides* or *P. infestans* were placed at the center of the KOF112-growing SCD plates. After incubation for 5 d, a large inhibition zone was formed between KOF112 and fungal mycelia compared with control culture. Bar, 1 cm. (**B**) Microscopic observation. Mycelial tips of growth-inhibited fungi co-incubated with KOF112 were swollen or ruptured compared with control culture. Bar, 100 μm. *Agro*, *Agrobacterium* sp. isolate CHB3 used as control isolate with no antifungal activity.

**Figure 2 plants-10-01815-f002:**
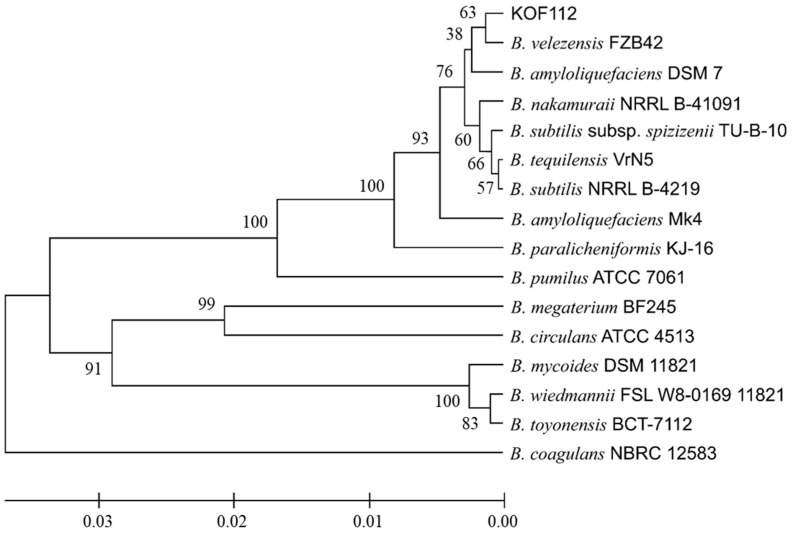
Phylogenetic analysis of 16S rDNA nucleotide sequence of KOF112 compared with those of *Bacillus* isolates. A phylogenetic tree was designed by means of the neighbor-joining (NJ) method using MEGA X program (bootstrap value with 1000 replicates). Bar indicates 1% band dissimilarity. Distance corresponds to the number of nucleotide substitutions per site. *B. velezensis* FZB42 (accession no. CP000560). *B. amyloliquefaciens* DSM7 (accession no. FN597644). *B. amyloliquefaciens* Mk4 (accession no. MT131178). *B. nakamuraii* NRRL B-41091 (accession no. KU836854). *B. subtilis* NRRL B-4219 (accession no. NR_116183). *B. tequilensis* VrN5 (accession no. LT986216). *B. subtilis* subsp. *spizizenjii* TU-B-10 (accession no. CP602905). *B. paralicheniformis* KJ-16 (accession no. KY694465). *B. pumilus* ATCC 7061 (accession no. AY876289). *B. megaterium* BF245 (accession no. MK491077). *B. circulans* ATCC 4513 (accession no. AY724690). *B. coagulans* NBRC 12583 (accession no. KX261624). *B. mycoides* DSM 11821 (accession no. AB021199). *B. wiedmannii* FSL W8-0169 11821 (accession no. KU198626). *B. toyonensis* BCT-7112 (accession no. CP006863).

**Figure 3 plants-10-01815-f003:**
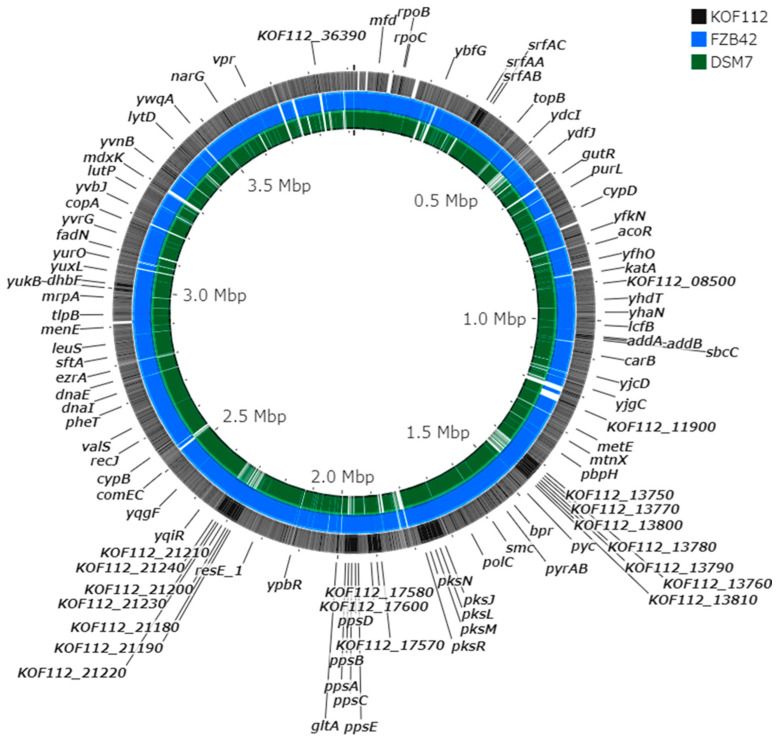
Genome comparison of KOF112, *B. velezensis* FZB42, and *B. amyloliquefaciens* DSM7. The whole genomes of KOF112 (outermost circle), FZB42 (middle circle), and DSM7 (innermost circle) were aligned using the CGView Server. The server used BLAST to compare the KOF112 genome sequence with FZB42 and DSM7 genome sequences. BLAST results and feature information for coding sequences were converted into a graphical map with black, blue, and green colors. Gene annotation of KOF112 coding sequences is partially shown in the figure.

**Figure 4 plants-10-01815-f004:**
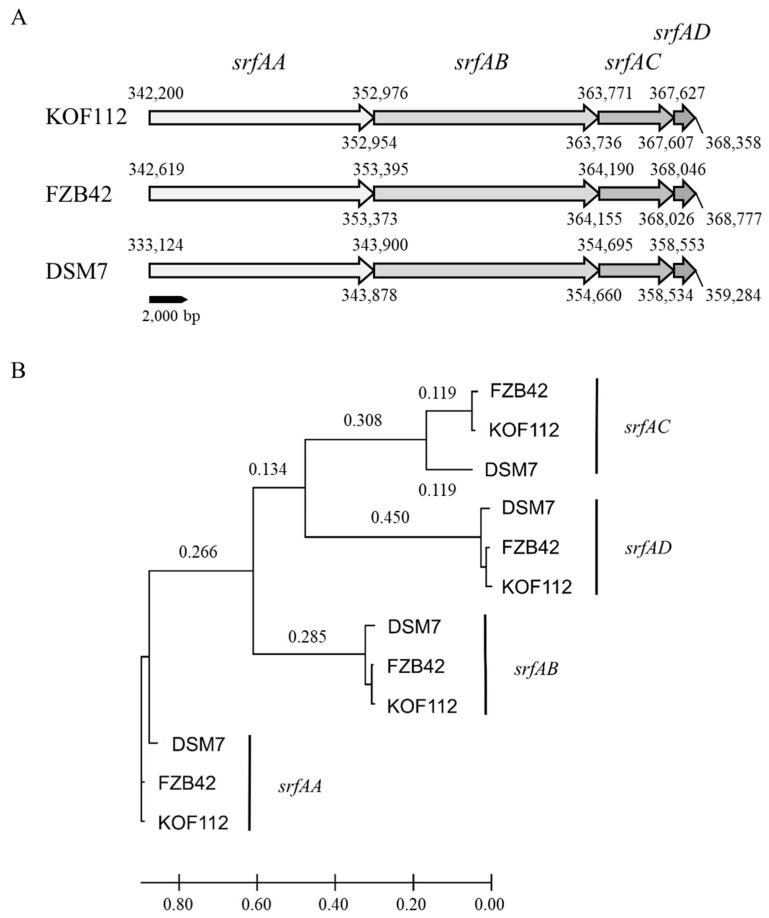
Comparison of surfactin biosynthetic gene clusters among KOF112, *B. velezensis* FZB42, and *B. amyloliquefaciens* DSM7. (**A**) Organization of surfactin biosynthesis genes. (**B**) Phylogenetic analysis of *srfAA*, *srfAB*, *srfAC*, and *srfAD* of KOF112 compared with those of FZB42 and DSM7. A phylogenetic tree was designed by means of the NJ method using MEGA X program (bootstrap value with 1000 replicates). Bar indicates 1% band dissimilarity. Distance corresponds to the number of nucleotide substitutions per site.

**Figure 5 plants-10-01815-f005:**
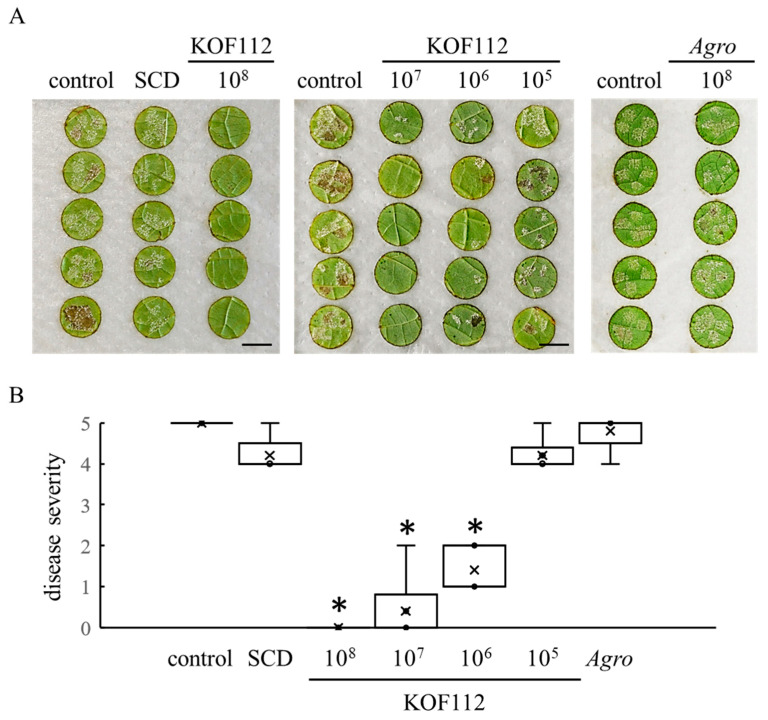
Suppression of grape downy mildew by KOF112. (**A**) Representative symptoms of grape downy mildew on leaf disks treated with KOF112 (1 × 10^8^, 1 × 10^7^, 1 × 10^6^ or 1 × 10^5^ cfu/mL) or *Agrobacterium* sp. isolate CHB3 (1 × 10^8^ cfu/mL). Bar, 1 cm. (**B**) Disease severity was evaluated as described in Materials and Methods. Crosses (×) indicate means of two independent experiments with five leaf disks. * *p* < 0.05 compared with control and SCD. Control, untreated. SCD, treated with 10% SCD. *Agro*, treated with *Agrobacterium* sp. isolate CHB3 used as control isolate with no antifungal activity.

**Figure 6 plants-10-01815-f006:**
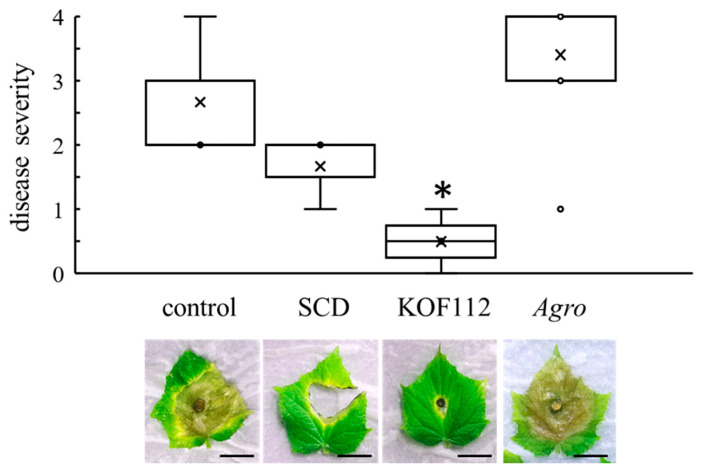
Suppression of cucumber gray mold by KOF112. Disease severity was evaluated as described in Materials and Methods. Crosses (×) indicate means of two independent experiments with three true leaves. * *p* < 0.05 compared with control and SCD. The representative symptom of cucumber gray mold on true leaves is shown below the graph. Control, untreated. SCD, treated with 10% SCD. KOF112, treated with 1 × 10^8^ cfu/mL KOF112. *Agro*, treated with 1 × 10^8^ cfu/mL *Agrobacterium* sp. isolate CHB3 used as control isolate with no antifungal activity. Bar, 2 cm.

**Figure 7 plants-10-01815-f007:**
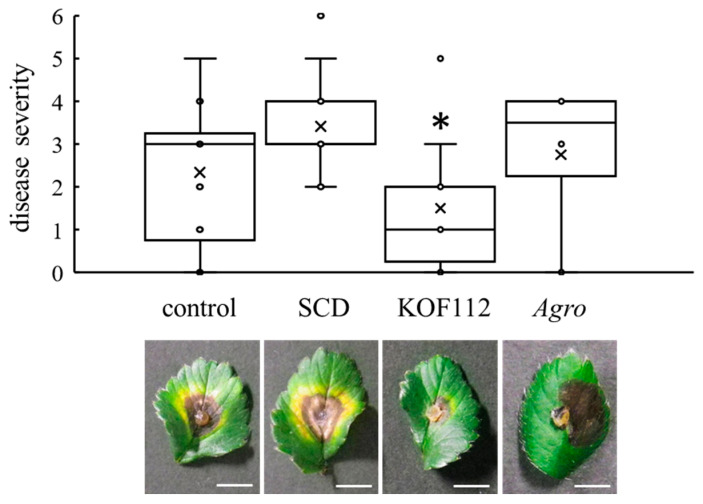
Suppression of strawberry anthracnose by KOF112. Disease severity was evaluated as described in Materials and Methods. Crosses (×) indicate means of four independent experiments with three leaves. * *p* < 0.05 compared with control and SCD. Each representative symptom of strawberry anthracnose on detached leaves is shown below the graph. Control, untreated. SCD, treated with 10% SCD. KOF112, treated with 1 × 10^8^ cfu/mL KOF112. *Agro*, treated with 1 × 10^8^ cfu/mL *Agrobacterium* sp. isolate CHB3, used as a control isolate with no antifungal activity. Bar, 1 cm.

**Figure 8 plants-10-01815-f008:**
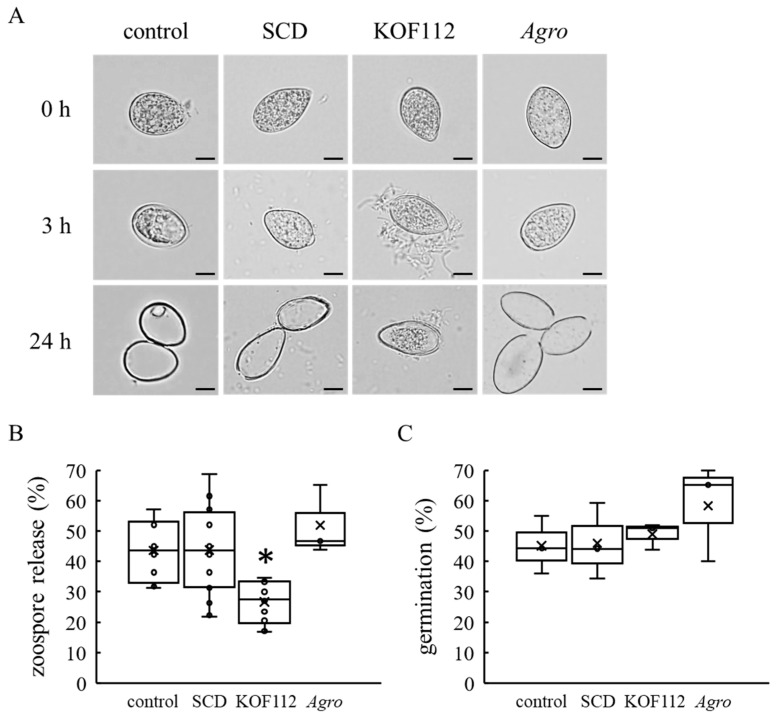
KOF112 inhibits zoospore release from zoosporangia but not zoospore germination. (**A**) Zoosporangia at 0, 3, and 24 h after preparation. Bar, 20 μm. (**B**) Zoospore release. The rates of zoospore release from zoosporangia were calculated as described in Materials and Methods. Crosses (×) indicate means of eight independent preparations. * *p* < 0.05 compared with control and SCD. (**C**) Zoospore gemination. The rates of zoospore germination were calculated as described in Materials and Methods. Crosses (×) indicate means of three independent preparations. Control, untreated. SCD, treated with 10% SCD. KOF112, treated with 1 × 10^8^ cfu/mL KOF112. Agro, treated with 1 × 10^8^ cfu/mL Agrobacterium sp. isolate CHB3, used as a control isolate with no antifungal activity.

**Figure 9 plants-10-01815-f009:**
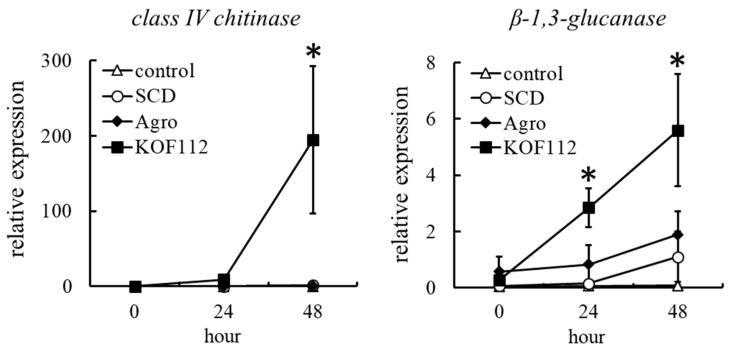
Transcription profiles of genes encoding class IV chitinase and β-1,3-glucanase in grape leaves treated with KOF112. Transcription levels of the genes in grape leaf disks 0, 24, and 48 h after KOF112 treatment were estimated by real-time RT-PCR. Data were calculated as gene expression relative to ubiquitin gene expression. Bars indicate means ± standard deviations of three independent experiments with three leaf disks. * *p* < 0.05 compared with control and SCD. Control, untreated. SCD, treated with 10% SCD. KOF112, treated with 1 × 10^8^ cfu/mL KOF112. *Agro*, treated with 1 × 10^8^ cfu/mL *Agrobacterium* sp. isolate CHB3, used as a control isolate with no antifungal activity.

## Data Availability

KOF112 genome was deposited in DDBJ/ENA/GenBank under accession numbers PRJDB11468 (BioProject), SAMD00293740 (BioSample), AP024603 (genome), AP024604 (plasmid), and DRA011804 (raw sequencing reads).

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
