# Peer review of "Isolation and Characterization of Endophyte Bacillus velezensis KOF112 from Grapevine Shoot Xylem as Biological Control Agent for Fungal Diseases"

_plants, 2021, doi:10.3390/plants10091815_

Round 1

Reviewer 1 Report

This manuscript by Hamaoka et al explored the biological control potential of an endophytic bacterial strain that was isolated from a grapevine xylem. Authors tested the in vitro and in vivo disease control potential of the isolate against fungal pathogens of grapevine. Authors present interesting results about the potential of the isolate to control diseases, as well as to elicit plant defense response. The genome and phylogeny part need more work or explanation to reach to the conclusions authors have reached. Writing is sub-optimal and needs meticulous revision.

Here are my specific comments.

 Line 83. Were these different bacteria? I feel like these are just that many colonies of bacteria.

 Figure 1 A. I think the inhibition zones are clear here, but pictures of control treatments (i.e. without the biocontrol bacteria, but just the buffer…) would have provided a better comparisons for readers who are not familiar with the dual assay. Also, although it is mentioned in the methods, it would be good to mention it in the figure legend that he bacteria are on the edge of plate and mycelia are at the centre. I am not familiar with the dual culture technique, so are the bacteria streaked in a tree like shape, or it is just how they grow?

Figure 1 B. Again, a control figure would help how do the not ruptured tips look. Just looking at these pictures, it’s hard to see what’s happening.

Figure 2. It is hard to read the tree as is. Please provide more information in the legend. Is it ML or NJ, how many bootstraps, rooted or not, what does the bar indicate etc. With the given BS values, it is hard to say if KOF112 is really a B. velezensis. For examples, B. subtilis isolates seem to cluster differently with greater BS values than between KOF112 and FZB42?

Lines 94-95. Why are these strains not included in the tree in Figure 2?

Line 98. I would suggest changing “impressive capacity’ to something quantifiable?

Figure 3 and lines 117-124. The figure is hard to read and not sure if depicts anything. It would be better to show similar regions with the same color and variable regions with different color. Maybe progressive mauve is useful? Also, if the genome comparisons were made, how were the ANI values? Do the whole genome comparison support the claim made earlier that KOF112 belongs to B. velezensis (given that authors mentioned they found significant differences between KOF112 and FZB42 in line 117-118).

Line 132. Please define what is SCD here.

Figure 4B. Based on Fig. 4A, the severity values of 10^5 seem to be lower than SCD, or at least should have a greater variance if the averages are same?

Line 245-253. “KOF112 is an unique producer for a cocktail of antibiotics”. I was not sure what was the basis for this claim? Does genome sequence suggest this? Antibiotics work against bacteria, right?

Line 415. How many biological replicates and how many times were these repeated. I think these are very important details to be included with every sections of the methods and maybe also in the figure legend.

Author Response

Point-by-point response to the comments (Reviewer #1):

Reviewers' comments:

  1. This manuscript by Hamaoka et al explored the biological control potential of an endophytic bacterial strain that was isolated from a grapevine xylem. Authors tested the in vitro and in vivo disease control potential of the isolate against fungal pathogens of grapevine. Authors present interesting results about the potential of the isolate to control diseases, as well as to elicit plant defense response. The genome and phylogeny part need more work or explanation to reach to the conclusions authors have reached. Writing is sub-optimal and needs meticulous revision.

Answer: Thank you very much for your advice to improve our manuscript. The manuscript was revised according to the reviewers’ suggestions. We hope that our revision is satisfactory and the revised manuscript is now acceptable for publication in the Plants.

  1. Line 83. Were these different bacteria?I feel like these are just that many colonies of bacteria.

Answer: As you pointed out, ‘colonies’ is right. We changed ‘endophytic bacteria’ to ‘colonies’ in the revised manuscript.

(See p. 2, line 84, please)

  1. Figure 1 A. I think the inhibition zones are clear here, but pictures of control treatments (i.e. without the biocontrol bacteria, but just the buffer…) would have provided a better comparisons for readers who are not familiar with the dual assay. Also, although it is mentioned in the methods, it would be good to mention it in the figure legend that he bacteria are on the edge of plate and mycelia are at the centre. I am not familiar with the dual culture technique, so are the bacteria streaked in a tree like shape, or it is just how they grow?

Answer: We have taken your suggestion and added pictures of control (fungal colonies without any treatment) in the revised Figure 1A. Also, simplified method for dual culture technique was included in the Figure legend.

        (See the revised Figure 1, please)

            KOF112 was streaked linearly on the edge of SCD plates and grew like tree.

  1. Figure 1 B. Again, a control figure would help how do the not ruptured tips look. Just looking at these pictures, it’s hard to see what’s happening.

Answer: We agreed with your suggestion, and added pictures of control (hyphal morphology without any treatment) in the revised Figure 1B

        (See the revised Figure 1B, please)

  1. Figure 2. It is hard to read the tree as is. Please provide more information in the legend. Is it ML or NJ, how many bootstraps, rooted or not, what does the bar indicate etc. With the given BS values, it is hard to say if KOF112 is really a B. velezensis. For examples, B. subtilis isolates seem to cluster differently with greater BS values than between KOF112 and FZB42?

Answer: We made the phylogenetic tree in Figure 2 easy to see in the revised Figure 2. Method for phylogenetic analysis was include in the legend of Figure 2. We agreed with your comment, and analyzed surfactin biosynthetic gene cluster as a non-ribosomal peptide synthase cluster among KOF112, B. velezensis FZB42 and B. amyloliquefaciens DSM7 in the revised Figure 4. Organization of srfAA, srfAB, srfAC, and srfAD perfectly matched among there isolates (Figure 4A). Phylogenetic analysis of each gene indicated that KOF112 surfactin biosynthetic genes formed a cluster with those of B. velezensis FZB42 (Figure 4B). With 16S rDNA analysis, we judged thatKOF112 is a strain of B. velezensis.

        (See the revised Figures 2 and 4, please)

  1. Lines 94-95. Why are these strains not included in the tree in Figure 2?

Answer: Since these strains have a perfect match with KOF112 16S rDNA sequenced, a cluster was formed with KOF112 in the phylogenetic tree. Therefore, other Bacillus species and B. velezensis FZB42, one of famous biocontrol agents, were analyzed with KOF112 in Figure 2.

  1. Line 98. I would suggest changing “impressive capacity’ to something quantifiable?

Answer: We are sorry that we could not think of anything quantifiable for impressive capacity . So, we changed ‘an impressive capacity’ to ‘a capacity’ in the revised manuscript.

(See p. 3, line 100, please)

  1. Figure 3 and lines 117-124. The figure is hard to read and not sure if depicts anything. It would be better to show similar regions with the same color and variable regions with different color. Maybe progressive mauve is useful? Also, if the genome comparisons were made, how were the ANI values? Do the whole genome comparison support the claim made earlier that KOF112 belongs to B. velezensis (given that authors mentioned they found significant differences between KOF112 and FZB42 in line 117-118).

Answer:  We revised Figure 3 for the readers to understand easily.

   (See the revised Figure 3, please)

      In addition, in the revised manuscript, we analyzed surfactin biosynthetic gene cluster as a non-ribosomal peptide synthase cluster among KOF112, B. velezensis FZB42 and B. amyloliquefaciens DSM7. Organization of srfAA, srfAB, srfAC, and srfAD perfectly matched among there isolates (Figure 4A). Phylogenetic analysis of each gene indicated that KOF112 surfactin biosynthetic genes formed a cluster with those of B. velezensis FZB42 (Figure 4B). With 16S rDNA analysis, we judged that KOF112 is a strain of B. velezensis.

  (See the revised Figure 4, please)

  1. Line 132. Please define what is SCD here.

Answer: Thank you for your advice. We included full name of SCD (soybean casein digest) here in the revised manuscript.

(See p. 7, line 166, please)

  1. Figure 4B. Based on Fig. 4A, the severity values of 10^5 seem to be lower than SCD, or at least should have a greater variance if the averages are same?

Answer: Statistical analysis (Dunnett’s multiple comparison test) did NOT detect any significant difference between 1 × 105 cfu/mL KOF112 and SCD treatments in Figure 4 (the revised Figure 5). Using our evaluation method (see below, please), Average and standard deviation of disease severity of SCD treatment were same as those of 1 × 105 cfu/mL KOF112 treatment.

  1. Line 245-253. “KOF112 is an unique producer for a cocktail of antibiotics”. I was not sure what was the basis for this claim? Does genome sequence suggest this? Antibiotics work against bacteria, right?

Answer: We have taken you suggestions, removed the sentence from the revised manuscript, and revised the part as follows:

        ‘Comparative genome analysis of KOF112 with FZB42 demonstrated that KOF112 might produce less antimicrobial peptides and polyketides than FZB42.’

        (See p. 12, lines 291-293, please)

  1. Line 415. How many biological replicates and how many times were these repeated. I think these are very important details to be included with every sections of the methods and maybe also in the figure legend.

Answer: Since the number of biological replicates and independent experiments were different among the experiments, these information were included in the sub-section of the Materials and methods section and the legend of each Figure.

Reviewer 2 Report

This review article entitled "Isolation and characterization of endophyte Bacillus velezensis KOF112 from grapevine shoot xylem as biological control agent for fungal diseases" is an attempt to show biocontrol activity of a newly isolated and characterized grapevine endophyte against three plant pathogens. However, the experimental design and description and the article structure lack crucial information:

General:

  1. The objectives are not clear, especially since there is no research question that this research addresses. For example, the reasons for exploring the endophytes are not well explained. Also, the authors mention that they extracted this potential biocontrol endophytic bacteria from grapevine; however, they did not mention which cultivar of grapevine (is it the Koshu cultivar that all the introduction is about?). Moreover, since this cultivar harbors these bacteria in their endophyte microbiota, does the cultivar show any resistance to pathogens?
  2. The background and introduction information is outdated and lacks important updates in biocontrol agents. For example, none of the research conducted by experts in this field, such as the group of Monica Höfte and Marc Ongena (only one review from 2008 was mentioned).
  3. There is no structure in the experimental design.
  4. The results are too abridged and missing crucial details.

Specifics: 

  1. In figure 1.B, the controls are missing. Therefore, I can't have confidence in the information it is supposed to deliver. No conclusions can be drawn.
  2. In the results section "2.2. KOF112 is a strain of Bacillus velezensis": 
    1. Bacillus velezensis should be in italics
    2. The authors performed the identification of an undefined isolate with only one gene, 16S rRNA. I find this not sufficient for the identification and classification of new bacterial isolates. Especially since the authors have already sequenced the genome of this bacteria species (as they mention in section 2.3), why didn't the authors do the phylogenetic analysis based on the whole genome, proteome, or at least based on the NPRS cluster as it is typically approached?
    3. In figure 2, the figure legend is missing crucial information, e.g., how many bootstraps replications were done, the method used for generating the phylogenetic tree and the phylogenetic analysis, what does the bar under the figure represent, ...?
  3. In the result section "2.3. Comparisons of KOF112 genome with B. velezensis FZB42 and B. amyloliquefaciens 115 DSM7 genomes":
  1. The authors mention that they did genome sequencing, assembly, and annotation; however, they have not reported here (or anywhere in the paper) any information about the genome (e.g., genome size, quality, completeness, tools...).
    1. When I searched the material and methods section for details about the sequencing, assembly, and annotation, the authors say nothing about it except that they followed the same protocol published in reference 37. However, when I checked reference 37 "Hamaoka, K.; Suzuki, S. Draft genome sequence of Bacillus velezensis KOF112, an antifungal endophytic isolate from a shoot 502 xylem of Japanese indigenous wine grape Vitis sp. cv. Koshu. Microbiol. Resour. Announc. 2021, under review." I found it still under review, and there is no link for the publication, even as a preprint article. Therefore, the quality of the genome assembly cannot be assessed.
    2. The authors claimed that they deposited the draft genome and the plasmid sequences of the KOF112 in DDBJ/ENA/GenBank under accession no. AP021846 and AP021847. However, when I checked this accession, I found genomes and plasmids of different bacteria, Pseudoalteromonas sp. A25. I found, however, a KOF112 assembly under accession GCA_018406485.1. Please provide accurate accessions for deposited genomes. In addition, the raw sequencing files need to be deposited as well. Are they available and under which accession precisely?
    3. The genome analysis is not clear. To be specific, what analysis was performed?
    4. It is not clear how they performed the NPRS cluster analysis and identification.
    5. In figure 3, I find it very vague. 
      1. I could only find one clp cluster, surfactin. The clp clusters should be highlighted in the figure.
      2. It is unclear what the comparison of the three genomes should demonstrate.
      3. Maybe an additional figure is required to show the presence/absence of the clp clusters/genes in comparison between the three other genomes.
  1. In results section "2.4. Biocontrol activity of KOF112 against downy mildew in grapevine:
  1. The SCD is an abbreviation that is first time mentioned with no description.
  2. Also, why the authors use SCD is not reported.
  1. In figure 4:
    1. In 4A, it appears that the SCD medium also affects the leaf discs since it shows clearly less disease severity. The same observation in figure 5. Therefore, one would ask if it contributes to the phenotype of bacterial activity of KOF112! 
    2. Figure 4B is an inappropriate data representation. It should show all the data points—the same comment for all the bar chart graphs in this manuscript.
  • In the results section "2.5. Biocontrol activity of KOF112 against gray mold in cucumber." 
    1. It is unclear why the authors did this assay on cucumber and strawberries since they are interested in solving the problem in the grapevine. It needs to be explained in that section.
    2. Crucial information about the assay is missing. For example, how many days after infection was the pathogen severity assay performed? The days after bacterial treatment? From the material and methods, it was unclear how the authors sprayed the bacteria and the plant leaves? Moreover, have the authors used any kind of detergent (e.g., tween or silvet, or triton-x) to keep the drops stable on the leaves or not?
    1. It is unclear why the authors used detached leaves in their assays since this is an unnatural infection method? With the amount of bacteria they used (108, which is very high) on unnatural infection systems, I suspect that the observation of less severity is due to the direct interaction of the bacteria with the pathogen and not through the plant itself. If the authors meant to draw this conclusion, this is missing as well.
  • In figure6, it is unclear if the significant difference of the KOF12 treatment is compared to the control or the SCD.
  • The experiments for figure 7 are missing the H2O controls. Therefore, the concluding results are not trustworthy because the SCD control shows differences from the H2O controls.
  • In results section "2.8. KOF112 induces plant defense response in grapevine":
  1. The authors did not mention the reason/s for choosing the two genes (class IV chitinase and β-1,3-glucanase) for qRT-PCR. 
  2. the qRT-PCR experiment is flawed for three reasons:
    1. The authors used an inappropriate reference gene (ß-actin), since it is responsive to pathogen infection (consult this reference for more information: DOI:10.1111/nph.16584).
    2. the authors based their analysis for gene expression on only one reference gene, which is not sufficient for accuracy
    3. Since SCD also induces changes in the phenotype, the authors should, as well, have tested the gene expression H2O control.
    4. primer efficiency has not be described/tested
    5. statistical analysis is unclear
  • The discussion is poor for many reasons:
  1. The authors should consult the latest reviews and research on biocontrol agents, cyclic lipopeptide, and plant-induced systemic resistance. To name a few, the group of Marc Ongena, Monica Hofta, and Corné Pieterse are experts in these fields for many years. However, none of their publications were consulted or cited here.
  2. The discussion is only a re-description of the results to justify the results obtained in this study
  3. results are not put in proper literature context
  4. there are no critical assessments of the data.

Author Response

Point-by-point response to the comments (Reviewer #2):

Reviewers' comments:

This review article entitled "Isolation and characterization of endophyte Bacillus velezensis KOF112 from grapevine shoot xylem as biological control agent for fungal diseases" is an attempt to show biocontrol activity of a newly isolated and characterized grapevine endophyte against three plant pathogens. However, the experimental design and description and the article structure lack crucial information:

Answer: Thank you very much for your advice to improve our manuscript. The manuscript was revised according to your suggestions. The following is our point-by-point response to the comments and detailing all changes made on the revised manuscript.

General:

  1. The objectives are not clear, especially since there is no research question that this research addresses. For example, the reasons for exploring the endophytes are not well explained. Also, the authors mention that they extracted this potential biocontrol endophytic bacteria from grapevine; however, they did not mention which cultivar of grapevine (is it the Koshu cultivar that all the introduction is about?). Moreover, since this cultivar harbors these bacteria in their endophyte microbiota, does the cultivar show any resistance to pathogens?

Answer: The objective of our study and the reason why we focused on endophytic bacteria was described in the Introduction section.

(See p. 2, lines 72-75, please)

We added the information about grapevine cultivars we used for isolation of endophytic bacteria in the revised Materials and methods section as follows:

       ‘Grapevines (Vitis. sp. cv. Koshu, V. vinifera cvs. Pinot Noir, Chardonnay, and Cabernet Sauvignon) were cultivated in the experimental vineyard of The Institute of Enology and Viticulture, University of Yamanashi, Yamanashi, Japan (latitude 35.680528, longitude 138.569268, elevation 250 m).’

(See p. 12, lines 322-325, please)

    Koshu is susceptible to grape downy mildew, but not to gray mold and ripe rot. We are pressing the searches for endophytic microbiota in Koshu xylem. Preliminary experiments demonstrates that KOF112 is not dominant in Koshu xylem.

  1. The background and introduction information is outdated and lacks important updates in biocontrol agents. For example, none of the research conducted by experts in this field, such as the group of Monica Höfte and Marc Ongena (only one review from 2008 was mentioned).

Answer: We present the information of background and introduction for our study with biocontrol to grapevine in viticulture in the Introduction section.

  1. There is no structure in the experimental design.

Answer: Thank you for your suggestion. In the revised manuscript, we added some experimental data and revised the overall flow of the manuscript. Please reads the revised manuscript again.

  1. The results are too abridged and missing crucial details.

Answer: According to you suggestions below, we added some experimental data and revised the Results and Discussion section in the revised manuscript. 

        (See the revised Results and Discussion sections, please)

Specifics:

  1. In figure 1.B, the controls are missing. Therefore, I can't have confidence in the information it is supposed to deliver. No conclusions can be drawn.

Answer: We added control’s pictures in Figures 1A and 1B.

        (See the revised Figure 1, please)

  1. In the results section "2.2. KOF112 is a strain of Bacillus velezensis":
  2. Bacillus velezensis should be in italics
  3. The authors performed the identification of an undefined isolate with only one gene, 16S rRNA. I find this not sufficient for the identification and classification of new bacterial isolates. Especially since the authors have already sequenced the genome of this bacteria species (as they mention in section 2.3), why didn't the authors do the phylogenetic analysis based on the whole genome, proteome, or at least based on the NPRS cluster as it is typically approached?
  4. In figure 2, the figure legend is missing crucial information, e.g., how many bootstraps replications were done, the method used for generating the phylogenetic tree and the phylogenetic analysis, what does the bar under the figure represent, ...?

Answer:  1. Since sub-title is italics, we judged that scientific name should not be italic.

  1. We analyzed surfactin biosynthetic gene cluster as a non-ribosomal peptide synthase cluster among KOF112, B. velezensis FZB42 and B. amyloliquefaciens DSM7 in the revised manuscript. Organization of srfAA, srfAB, srfAC, and srfAD perfectly matched among there isolates (Figure 4A). Phylogenetic analysis of each gene indicated that KOF112 surfactin biosynthetic genes formed a cluster with those of B. velezensis FZB42 (Figure 4B). With 16S rDNA analysis, the results suggest that KOF112 is a strain of B. velezensis.

           (See the revised Figure 4, please)

  1. We added the methods for the phylogenetic analysis in the Figure legend of Figure 2 as follows:

 ‘A phylogenetic tree was designed by means of the Neighbor-Joining (NJ) method using MEGA X program (bootstrap value with 1000 replicates). Bar indicates 1% band dissimilarity. Distance corresponds to the number of nucleotide substitutions per site.’

Also, B. amyloliquefaciens DSM7 was included in the analysis.

   (See the revised Figure 2, please)

  1. In the result section "2.3. Comparisons of KOF112 genome with B. velezensis FZB42 and B. amyloliquefaciens 115 DSM7 genomes":
  2. The authors mention that they did genome sequencing, assembly, and annotation; however, they have not reported here (or anywhere in the paper) any information about the genome (e.g., genome size, quality, completeness, tools...).

Answer: We published genome announcement of KOF112 in Microbiology Resource Announcement (Accepted on June 21, 2021 and in print). In the revised manuscript, brief information of genome analysis was added in the revised Materials and methods section.

               (See p. 12, lines 362-367, please)

                 Also, we added the information of KOF112 genome (size and annotation) in the revised Results section.

               (See p. 3, lines 102-104, please)

  1. When I searched the material and methods section for details about the sequencing, assembly, and annotation, the authors say nothing about it except that they followed the same protocol published in reference 37. However, when I checked reference 37 "Hamaoka, K.; Suzuki, S. Draft genome sequence of Bacillus velezensis KOF112, an antifungal endophytic isolate from a shoot 502 xylem of Japanese indigenous wine grape Vitis sp. cv. Koshu. Microbiol. Resour. Announc. 2021, under review." I found it still under review, and there is no link for the publication, even as a preprint article. Therefore, the quality of the genome assembly cannot be assessed.

Answer: Genome announcement of KOF112 was accepted in Microbiology Resource Announcement on June 21, 2021, and just in print. We will present DOI number as soon as possible.

  1. The authors claimed that they deposited the draft genome and the plasmid sequences of the KOF112 in DDBJ/ENA/GenBank under accession no. AP021846 and AP021847. However, when I checked this accession, I found genomes and plasmids of different bacteria, Pseudoalteromonas sp. A25. I found, however, a KOF112 assembly under accession GCA_018406485.1. Please provide accurate accessions for deposited genomes. In addition, the raw sequencing files need to be deposited as well. Are they available and under which accession precisely?

Answer:  We are sorry for the mistake. This is perfectly our mistake. AP021846 and AP021847 were wrong. AP024603 (genome) and AP024604 (plasmid) were right. We corrected the accession numbers in the revised manuscript.

              (See p. 13, lines 359-369, please)

  1. The genome analysis is not clear. To be specific, what analysis was performed?

Answer:  We added the information of genome comparison using the CGView Server in the revised legend of Figure 3.

              (See the revised Figure 3, please)

  1. It is not clear how they performed the NPRS cluster analysis and identification.

Answer:  NPRS (non-ribosomal peptide synthase) analysis was performed by BLAST homology research using the CGView Server. In the revised manuscript, we analyzed surfactin biosynthetic gene cluster as a NPRS cluster among KOF112, B. velezensis FZB42 and B. amyloliquefaciens DSM7 in the revised Figure 4. Organization of srfAA, srfAB, srfAC, and srfAD perfectly matched among there isolates (Figure 4A). Phylogenetic analysis of each gene indicated that KOF112 surfactin biosynthetic genes formed a cluster with those of B. velezensis FZB42 (Figure 4B).

         (See the revised Figure 4, please)

  1. In figure 3, I find it very vague.
  2. I could only find one clp cluster, surfactin. The clp clusters should be highlighted in the figure.
  3. It is unclear what the comparison of the three genomes should demonstrate.
  4. Maybe an additional figure is required to show the presence/absence of the clp clusters/genes in comparison between the three other genomes.

Answer:  6-1. Thank you for your advice. We created new Figure (Figure 4 in the revised manuscript) for surfactin biosynthetic gene clusteranalysis.

             (See the revised Figure 4, please)

6-2. The CGView Server, that we used in our study, used BLAST to compare KOF112 genome sequence to FZB42 and DSM7 genome sequences. The BLAST results and feature information for coding sequences were converted to a graphical map with black, blue, and green in the revised manuscript. This information for genome comparison was added in the legend of the revised Figure 3.

    (See the revised Figure 3, please)

6-3. We analyzed surfactin biosynthetic gene cluster as a NPRS cluster among KOF112, B. velezensis FZB42 and B. amyloliquefaciens DSM7 in the revised Figure 4.

(See the revised Figure 4, please)

Other NPRS were described in the text of the revised manuscript.

    (See p. 3, lines 107-111, please)

  1. In results section "2.4. Biocontrol activity of KOF112 against downy mildew in grapevine:
  2. The SCD is an abbreviation that is first time mentioned with no description.
  3. Also, why the authors use SCD is not reported.

Answer: 1. We showed the abbreviation of SCD in that place.

          (See p. 7, line 166, please)

  1.   We added the explanation why we used SCD in the experiment as follows:

                 ‘Since KOF112 inoculum contained 10% soybean casein digest (SCD) medium, 10% SCD medium was used as a control for assays of biocontrol activity of KOF112.’

(See p. 7, lines 166-167, please)

  1. In figure 4:
  2. In 4A, it appears that the SCD medium also affects the leaf discs since it shows clearly less disease severity. The same observation in figure 5. Therefore, one would ask if it contributes to the phenotype of bacterial activity of KOF112!
  3. Figure 4B is an inappropriate data representation. It should show all the data points—the same comment for all the bar chart graphs in this manuscript.

Answer: 1. Statistical analysis (Dunnett’s multiple comparison test) did NOT detect any significant difference between control (untreated) and SCD treatment in Figure 4, 5, and 6 (the revised Figure 5, 6, and 7, respectively). Therefore, we can’t describe any effect of SCD treatment on disease severities.  

  1.         Scatter plot and/or box plot are not suitable for our graphs, because the graphs were created from less than 10 data points. Unfortunately, we hope that We hope that our graphs are satisfactory for publication in the Plants.

  1. In the results section "2.5. Biocontrol activity of KOF112 against gray mold in cucumber."
  2. It is unclear why the authors did this assay on cucumber and strawberries since they are interested in solving the problem in the grapevine. It needs to be explained in that section.
  3. Crucial information about the assay is missing. For example, how many days after infection was the pathogen severity assay performed? The days after bacterial treatment? From the material and methods, it was unclear how the authors sprayed the bacteria and the plant leaves? Moreover, have the authors used any kind of detergent (e.g., tween or silvet, or triton-x) to keep the drops stable on the leaves or not?
  4. It is unclear why the authors used detached leaves in their assays since this is an unnatural infection method? With the amount of bacteria they used (108, which is very high) on unnatural infection systems, I suspect that the observation of less severity is due to the direct interaction of the bacteria with the pathogen and not through the plant itself. If the authors meant to draw this conclusion, this is missing as well.

Answer: 1. The reason why we used cucumber-B. cinerea and strawberry-C. gloeosporioides pathosystems was described in the Materials and methods of former manuscript. In the revised manuscript, the explanations have been moved in the Results section.

           (See p. 8, lines 183-184 and p. 9, lines 196-197, please)

  1. The experimental conditions were described in the Materials and methods section. In the legends of Figures, we included comments in each Figure legend as follows:

          ‘Disease severity was evaluated as described in Materials and Methods.’

          (See the revised Figure legends of Figures 5, 6, and 7, please)

              By the way, we did NOT use any detergent in the inoculation tests, since we used detached leaves and leaf disks to keep the drops of inoculum stable.

  1. We used detached leaves and leaf disks to keep perfectly KOF112 on leaves. 1 × 108 cfu/mL of biological control agents is not too much, since commercial biological fungicides (Bacillus) recommend 1 × 108 cfu/mL of spores as a formula.

     In our manuscript, we demonstrated that KOF112 showed biocontrol activities toward gray mold caused by B. cinerea, anthracnose by C. gloeosporioides, and downy mildew by P. viticola. The KOF112-inhibited mycelial tips were swollen or ruptured, suggesting that KOF112 produces antifungal materials. In addition, KOF112 induced plant defense response in grapevine. Recently, enhancing plant defense response by antifungal metabolites produced by biological control agents is likely one of the mechanisms responsible for protecting plants. Therefore, we described the relationship between direct antifungal activity and induction of plant defense response by antifungal materials.

         (See the revised Discussion section, please)

  1. In figure6, it is unclear if the significant difference of the KOF12 treatment is compared to the control or the SCD.

Answer: Statistical analysis detected the significant difference between KOF112 treatment and control (untreated) or SCD treatment. An asterisk above the column (KOF112) indicated statistically significant difference (p < 0.05).

(See the revised Figure 7, please)

  1. The experiments for figure 7 are missing the H2O controls. Therefore, the concluding results are not trustworthy because the SCD control shows differences from the H2O controls.

Answer: According to your suggestion, we added untreated control in the Figures 7 and 8 (the revised Figures 8 and 9, respectively).

        (See the revised Figures 8 and 9, please)

  1. In results section "2.8. KOF112 induces plant defense response in grapevine":
  2. The authors did not mention the reason/s for choosing the two genes (class IV chitinase and β-1,3-glucanase) for qRT-PCR.
  3. the qRT-PCR experiment is flawed for three reasons:
  4. The authors used an inappropriate reference gene (ß-actin), since it is responsive to pathogen infection (consult this reference for more information: DOI:10.1111/nph.16584).
  5. the authors based their analysis for gene expression on only one reference gene, which is not sufficient for accuracy
  6. Since SCD also induces changes in the phenotype, the authors should, as well, have tested the gene expression H2O control.
  7. primer efficiency has not be described/tested
  8. statistical analysis is unclear

Answer: 1. As you pointed out, we added the explanation why we selected the two genes (class IV chitinase and β-1,3-glucanase) in the experiment as follows:

‘We selected genes encoding class IV chitinase and β-1,3-glucanase as indicators of plant defense response, since gene expression chitinase and β-1,3-glucanase was induced through jasmonic acid and salicylic acid-dependent defense pathways, respectively.’

(See p. 9-10, lines 231-233, please)

        2-1.  β-Actin were used as the reference gene for the normalization of each gene expression in grapevine, since actin was stable in the P. viticola-grapevine interaction (FIGUEIREDO et al., 2012; POLESANI et al., 2010).

                 Figueiredo A., Monteiro F., Fortes A.M., Bonow-Rex M., Zyprian E., Sousa L., Pais M.S. (2012): Cultivar-specific kinetics of gene induction during downy mildew early infection in grapevine. Functional & Integrative Genomics, 12: 379-386.

                  Polesani M., Bortesi L., Ferrarini A., Zamboni A., Fasoli M., Zadra C., Lovato A., Pezzotti M., Delledonne M., Polverari A. (2010): General and species-specific transcriptional responses to downy mildew infection in a susceptible (Vitis vinifera) and a resistant (V. riparia) grapevine species. BMC Genomics, 11: 117.

               Since KOF112 is not a pathogen in grapevine, actin remodelling might not be induced by KOF112 inoculation.

2-2.  As mentioned above, actin gene expression was stable in grapevine. Therefore, we selected β-Actin as a reference gene for real-time RT-PCR analysis.

2-3. According to your suggestion, we added untreated control in the Figure 8 (the revised Figure 9). There were any significant differences between untreated control and SCD treatment at any time points.

    (See the revised Figure 9, please)

2-4. Same primer sets were used in our published papers, and there have been no problems so far.

        For example,

           Aoki, Y., Usujima, A. and Suzuki, S. (2021) High night temperature promotes downy mildew in grapevine via attenuating plant defense response and enhancing early Plasmopara viticola infection. Plant Protection Science 57:21-30.

Ishiai, S., Kondo, H., Hattori, T., Mikami, M., Aoki, Y., Enoki, S. and Suzuki, S. (2016) Hordenine is responsible for plant defense response through jasmonate-dependent defense pathway. Physiological and Molecular Plant Pathology 96:94-100.

 2-5. Data are presented as means ± standard deviations of three independent experiments with three leaf disks. Statistical analysis was performed by the Dunnett’s multiple comparison test. Asterisks indicated statistically significant difference (p < 0.05) against untreated control and SCD treatments.

     (See the revised Figure 9, please)

  1. The discussion is poor for many reasons:
  2. The authors should consult the latest reviews and research on biocontrol agents, cyclic lipopeptide, and plant-induced systemic resistance. To name a few, the group of Marc Ongena, Monica Hofta, and Corné Pieterse are experts in these fields for many years. However, none of their publications were consulted or cited here.
  3. The discussion is only a re-description of the results to justify the results obtained in this study
  4. results are not put in proper literature context
  5. there are no critical assessments of the data.

Answer: 1. We added deeper discussions using the following references, published in 2019-2021 years, in the revised manuscript.

Bruisson S, Zufferey M, L'Haridon F, Trutmann E, Anand A, Dutartre A, De Vrieze M, Weisskopf L. Endophytes and Epiphytes From the Grapevine Leaf Microbiome as Potential Biocontrol Agents Against Phytopathogens. Front Microbiol. 2019 Nov 29;10:2726. doi: 10.3389/fmicb.2019.02726. PMID: 31849878; PMCID: PMC6895011.

Li Y, Héloir MC, Zhang X, Geissler M, Trouvelot S, Jacquens L, Henkel M, Su X, Fang X, Wang Q, Adrian M. Surfactin and fengycin contribute to the protection of a Bacillus subtilis strain against grape downy mildew by both direct effect and defence stimulation. Mol Plant Pathol. 2019 Aug;20(8):1037-1050. doi: 10.1111/mpp.12809. Epub 2019 May 18. PMID: 31104350; PMCID: PMC6640177.

Leal C, Fontaine F, Aziz A, Egas C, Clément C, Trotel-Aziz P. Genome sequence analysis of the beneficial Bacillus subtilis PTA-271 isolated from a Vitis vinifera (cv. Chardonnay) rhizospheric soil: assets for sustainable biocontrol. Environ Microbiome. 2021 Jan 29;16(1):3. doi: 10.1186/s40793-021-00372-3. PMID: 33902737; PMCID: PMC8067347.

                  (See the revised Discussion section, please)

  1. We agreed with your suggestions, and revised the Discussion section in the revised manuscript.

(See the revised Discussion section, please)

  1. We revised the Discussion section with new four references.

(See the revised Discussion section, please)

  1. We revised the Discussion section with data-based discussions.

(See the revised Discussion section, please)

Reviewer 3 Report

The article reports the isolation of a Bacillus velezensis strain from internal tissues of grapevine, its genomic properties, and its performance as an in vitro competitor of grapevine phytopathogenic fungi, as a biological control agent of these fungi in plant tissues, its impact on zoospore release from sporangia, and finally its ability to induce expression of plant defense-related marker genes.

The manuscript is well written in general, data is presented in a proper way, and most of the results are solid.

One main concern is the interpretation of the results related to the ascribed biocontrol activity of isolate KOF122 in those assays with plant tissues. The in vitro dual culture tests suggest KOF122 produce diffusible compounds that inhibit the growth of the three tested fungi. The co-inoculation assays in plant tissues show that there is a dose-dependent protection associated with the presence of KOF122. However, this does not mean that this protection is due to antibiotic(s) produced by KOF122, and moreover, it does not mean that the protective activity is specific for strain KOF122. The protection is achieved with a large concentration of bacteria applied to plant tissues (10e8 cells). In this scenario, could competition for niche be discarded as the cause for the protection? What if an unrelated bacterium lacking antifungal BGCs is used instead, and applied to plant tissues at this concentration? It would be important to rule out a biocontrol-independent competitive mechanism. For this reason, I strongly suggest testing an unrelated saprophytic bacterium as control, to reinforce (or not) the relevance of direct antibiosis and/or any other biocontrol mechanism specific for strain KOF122.

Section 2.8.: Again, I think it is very important to include another bacterium to compare the plant response to KOF112. It would add a strong support to the proposed use of KOF112, if the defense response is specifically induced by KOF112 only. 

Figure 1: this image requires inclusion of panels from untreated fungi showing the intact structures of the mycelial filaments and tips. Otherwise it is difficult to ascertain the claimed damage.

Lines 262-264: ".... However, so far, we cannot demonstrate any positive results of KOF112 colonization in grapevine." I wonder which would be the reason of this statement. What does it mean? That KOF122 cannot be recovered from inoculated leaves? Or that KOF112 cannot be detected by molecular methods and/or by microscopy?

Section 2.3., comparative genomic analysis of strain KOF122: I think this section would benefit from an expansion on the overall genomic properties of KOF112, and its comparison with the BGC wealth of strain FZB42 (see Fan et al 2018; Ruckert et al 2011) . Figure 3: why not marking the BGCs related to biocontrol in this figure, to highlight similarities & differences?

Some missing references to related works that should be discussed:

Bruisson S, Zufferey M, L'Haridon F, Trutmann E, Anand A, Dutartre A, De Vrieze M, Weisskopf L. Endophytes and Epiphytes From the Grapevine Leaf Microbiome as Potential Biocontrol Agents Against Phytopathogens. Front Microbiol. 2019 Nov 29;10:2726. doi: 10.3389/fmicb.2019.02726. PMID: 31849878; PMCID: PMC6895011.

Li Y, Héloir MC, Zhang X, Geissler M, Trouvelot S, Jacquens L, Henkel M, Su X, Fang X, Wang Q, Adrian M. Surfactin and fengycin contribute to the protection of a Bacillus subtilis strain against grape downy mildew by both direct effect and defence stimulation. Mol Plant Pathol. 2019 Aug;20(8):1037-1050. doi: 10.1111/mpp.12809. Epub 2019 May 18. PMID: 31104350; PMCID: PMC6640177.

Leal C, Fontaine F, Aziz A, Egas C, Clément C, Trotel-Aziz P. Genome sequence analysis of the beneficial Bacillus subtilis PTA-271 isolated from a Vitis vinifera (cv. Chardonnay) rhizospheric soil: assets for sustainable biocontrol. Environ Microbiome. 2021 Jan 29;16(1):3. doi: 10.1186/s40793-021-00372-3. PMID: 33902737; PMCID: PMC8067347.

Author Response

Point-by-point response to the comments (Reviewer #3):

Reviewers' comments:

  1. The article reports the isolation of a Bacillus velezensis strain from internal tissues of grapevine, its genomic properties, and its performance as an in vitro competitor of grapevine phytopathogenic fungi, as a biological control agent of these fungi in plant tissues, its impact on zoospore release from sporangia, and finally its ability to induce expression of plant defense-related marker genes.

The manuscript is well written in general, data is presented in a proper way, and most of the results are solid.

Answer: Thank you very much for your advice to improve our manuscript. The manuscript was revised according to your suggestions. We hope that our revision is satisfactory and the revised manuscript is now acceptable for publication in the Plants.

  1. One main concern is the interpretation of the results related to the ascribed biocontrol activity of isolate KOF122 in those assays with plant tissues. The in vitro dual culture tests suggest KOF122 produce diffusible compounds that inhibit the growth of the three tested fungi. The co-inoculation assays in plant tissues show that there is a dose-dependent protection associated with the presence of KOF122. However, this does not mean that this protection is due to antibiotic(s) produced by KOF122, and moreover, it does not mean that the protective activity is specific for strain KOF122. The protection is achieved with a large concentration of bacteria applied to plant tissues (10e8 cells). In this scenario, could competition for niche be discarded as the cause for the protection? What if an unrelated bacterium lacking antifungal BGCs is used instead, and applied to plant tissues at this concentration? It would be important to rule out a biocontrol-independent competitive mechanism. For this reason, I strongly suggest testing an unrelated saprophytic bacterium as control, to reinforce (or not) the relevance of direct antibiosis and/or any other biocontrol mechanism specific for strain KOF122.

Answer: We judged that 1×108 cfu/mL of biological control agents is not too much, since commercial biological fungicides (Bacillus) recommend 1 × 108 cfu/mL of spores as a formula.

We didn’t use unrelated saprophytic bacteria as control in the revised manuscript.  The saprophytic bacteria are not suitable as control in the present study, since the saprophytic bacteria also produce and secrete a number of secondary metabolites. The experiments with the saprophytic bacteria will lose the original purpose.

              In our manuscript, we demonstrated that KOF112 showed biocontrol activities toward gray mold caused by B. cinerea, anthracnose by C. gloeosporioides, and downy mildew by P. viticola. The KOF112-inhibited mycelial tips were swollen or ruptured, suggesting that KOF112 produces antifungal materials. In addition, KOF112 induced plant defense response in grapevine. Recently, enhancing plant defense response by antifungal metabolites produced by biological control agents is likely one of the mechanisms responsible for protecting plants. Therefore, we described the relationship between direct antifungal activity and induction of plant defense response by antifungal materials in the revised Discussion section.

         (See the revised Discussion section, please)

  1. Section 2.8.: Again, I think it is very important to include another bacterium to compare the plant response to KOF112. It would add a strong support to the proposed use of KOF112, if the defense response is specifically induced by KOF112 only.

Answer: Answer is same as comment #2 above. Again, w didn’t use unrelated saprophytic bacteria as control in the revised manuscript.  The saprophytic bacteria are not suitable as control in the present study, since the saprophytic bacteria also produce and secrete a number of secondary metabolites. The experiments with the saprophytic bacteria will lose the original purpose.

              In our manuscript, we demonstrated that KOF112 showed biocontrol activities toward gray mold caused by B. cinerea, anthracnose by C. gloeosporioides, and downy mildew by P. viticola. The KOF112-inhibited mycelial tips were swollen or ruptured, suggesting that KOF112 produces antifungal materials. In addition, KOF112 induced plant defense response in grapevine. Recently, enhancing plant defense response by antifungal metabolites produced by biological control agents is likely one of the mechanisms responsible for protecting plants. Therefore, we described the relationship between direct antifungal activity and induction of plant defense response by antifungal materials in the revised Discussion section.

       (See the revised Discussion section, please) 

  1. Figure 1: this image requires inclusion of panels from untreated fungi showing the intact structures of the mycelial filaments and tips. Otherwise it is difficult to ascertain the claimed damage.

Answer: We have taken your suggestion, and added pictures of control (fungal colonies and hyphal morphologies without any treatment) in the revised Figures 1A and 1B.

        (See the revised Figure 1, please)

  1. Lines 262-264: ".... However, so far, we cannot demonstrate any positive results of KOF112 colonization in grapevine." I wonder which would be the reason of this statement. What does it mean? That KOF122 cannot be recovered from inoculated leaves? Or that KOF112 cannot be detected by molecular methods and/or by microscopy?

Answer: According to you suggestion, we removed the sentence from the revised manuscript.

       (See p. 12, lines 314-315, please)

  1. Section 2.3., comparative genomic analysis of strain KOF122: I think this section would benefit from an expansion on the overall genomic properties of KOF112, and its comparison with the BGC wealth of strain FZB42 (see Fan et al 2018; Ruckert et al 2011) . Figure 3: why not marking the BGCs related to biocontrol in this figure, to highlight similarities & differences?

Answer:  Thank you for your suggestion. We analyzed surfactin biosynthetic gene cluster as a non-ribosomal peptide synthase cluster among KOF112, B. velezensis FZB42 and B. amyloliquefaciens DSM7 in the revised manuscript. Organization of srfAA, srfAB, srfAC, and srfAD perfectly matched among there isolates (the revised Figure 4A). Phylogenetic analysis of each gene indicated that KOF112 surfactin biosynthetic genes formed a cluster with those of B. velezensis FZB42 (Figure 4B). With 16S rDNA analysis, the results suggest that KOF112 is a strain of B. velezensis.

          (See the revised Figure 4, please)

  1. Some missing references to related works that should be discussed:

Bruisson S, Zufferey M, L'Haridon F, Trutmann E, Anand A, Dutartre A, De Vrieze M, Weisskopf L. Endophytes and Epiphytes From the Grapevine Leaf Microbiome as Potential Biocontrol Agents Against Phytopathogens. Front Microbiol. 2019 Nov 29;10:2726. doi: 10.3389/fmicb.2019.02726. PMID: 31849878; PMCID: PMC6895011.

Li Y, Héloir MC, Zhang X, Geissler M, Trouvelot S, Jacquens L, Henkel M, Su X, Fang X, Wang Q, Adrian M. Surfactin and fengycin contribute to the protection of a Bacillus subtilis strain against grape downy mildew by both direct effect and defence stimulation. Mol Plant Pathol. 2019 Aug;20(8):1037-1050. doi: 10.1111/mpp.12809. Epub 2019 May 18. PMID: 31104350; PMCID: PMC6640177.

Leal C, Fontaine F, Aziz A, Egas C, Clément C, Trotel-Aziz P. Genome sequence analysis of the beneficial Bacillus subtilis PTA-271 isolated from a Vitis vinifera (cv. Chardonnay) rhizospheric soil: assets for sustainable biocontrol. Environ Microbiome. 2021 Jan 29;16(1):3. doi: 10.1186/s40793-021-00372-3. PMID: 33902737; PMCID: PMC8067347.

Answer: We have taken your suggestion, and revised the Discussion section with the references you suggested.

(See the revised Discussion section, please)

Round 2

Reviewer 2 Report

Additional analysis has been done and substantial improvements have been made in this revision. Many of my previous questions/concerns have been addressed. There are however important issues that have not been properly addressed:

  1. My previous comment: The background and introduction information is outdated and lacks important updates in biocontrol agents. For example, none of the research conducted by experts in this field, such as the group of Monica Höfte and Marc Ongena (only one review from 2008 was mentioned).

Authors’ answer: We present the information of background and introduction for our study with biocontrol to grapevine in viticulture in the Introduction section.

There are several studies about biocontrol agents in grapevine. Here are just two examples (Heyman, Lisa, et al. “Potential of Pseudomonas Cyclic Lipopeptides to Control Downy Mildew in Grapevine by Induced Resistance and Direct Antagonism.” MOLECULAR PLANT-MICROBE INTERACTIONS, vol. 32, no. 10, Amer Phytopathological Soc, 2019, pp. 95–95; “Surfactin and fengycin contribute to the protection of a Bacillus subtilis strain against grape downy mildew by both direct effect and defence stimulation” https://doi.org/10.1111/mpp.12809). It is crucial to discuss similar research in the field; otherwise, this gives an impression that the authors are intentionally omitting these studies to claim novelty. In addition, the background is too narrow by focusing only on describing the problem and naming the biofungal agents that are currently used and not talking about the known molecular background of these agents, such as the cyclic lipopeptides or antifungal compounds. Especially since the results highlighted the presence of the cyclic lipopeptides in the identified strain. It is crucial to provide background about these secondary metabolites that is a main feature of these beneficial bacteria (especially Bacillus velezensis). Cyclic lipopeptides are known to increase plant resistance and have antimicrobial activities, which is main part of your study.

  1. My previous comment: Figure 4B is an inappropriate data representation. It should show all the data points—the same comment for all the bar chart graphs in this manuscript.

Answer: Scatter plot and/or box plot are not suitable for our graphs, because the graphs were created from less than 10 data points. Unfortunately, we hope that We hope that our graphs are satisfactory for publication in the Plants.

This is not true. You can make boxplots with as few as three datapoints (although at least five should be used). Also, even on a par-chart graph you can easily add the datapoints. It is absolutely required to plot the datapoints since it shows where most of your observation lie on the range of the replicates. Otherwise, the graph oversimplifies the findings as it only shows the average, which is often skewed by outliers and non-normal distribution of the data. Therefore, this data representation can be misleading and omits crucial information to judge and interpret the data. Refer to e.g. 10.1371/journal.pbio.1002128 for clarification.

  1. My previous comment: It is unclear why the authors used detached leaves in their assays since this is an unnatural infection method? With the amount of bacteria they used (108, which is very high) on unnatural infection systems, I suspect that the observation of less severity is due to the direct interaction of the bacteria with the pathogen and not through the plant itself. If the authors meant to draw this conclusion, this is missing as well.

Authors’ answer: We used detached leaves and leaf disks to keep perfectly KOF112 on leaves. 1 × 108 cfu/mL of biological control agents is not too much, since commercial biological fungicides (Bacillus) recommend 1 × 108 cfu/mL of spores as a formula.

In our manuscript, we demonstrated that KOF112 showed biocontrol activities toward gray mold caused by B. cinerea, anthracnose by C. gloeosporioides, and downy mildew by P. viticola. The KOF112-inhibited mycelial tips were swollen or ruptured, suggesting that KOF112 produces antifungal materials. In addition, KOF112 induced plant defense response in grapevine. Recently, enhancing plant defense response by antifungal metabolites produced by biological control agents is likely one of the mechanisms responsible for protecting plants. Therefore, we described the relationship between direct antifungal activity and induction of plant defense response by antifungal materials.

Your answer confirmed my concern that the phenotype that you see on the plant leaf discs is not due to increasing plant resistance but rather direct interaction/antagonism of the bacteria with the fungus. Particularly since you showed that the bacteria have antifungal activity in vitro and appear to harbor antifungal compounds against these fungi.

In addition, you use the same amount of bacteria (1 × 108 cfu/mL) in an unnatural setup (detached leaves) as is recommended for using the commercial biological fungicide in the field. I am concerned that the phenotype you are observing does not support the interpretation of increased plant resistance.

  1. My previous comment: The authors used an inappropriate reference gene (ß-actin), since it is responsive to pathogen infection (consult this reference for more information: DOI:10.1111/nph.16584).

Authors’ answer: β-Actin were used as the reference gene for the normalization of each gene expression in grapevine, since actin was stable in the P. viticola-grapevine interaction (FIGUEIREDO et al., 2012; POLESANI et al., 2010).

                 Figueiredo A., Monteiro F., Fortes A.M., Bonow-Rex M., Zyprian E., Sousa L., Pais M.S. (2012): Cultivar-specific kinetics of gene induction during downy mildew early infection in grapevine. Functional & Integrative Genomics, 12: 379-386.

                  Polesani M., Bortesi L., Ferrarini A., Zamboni A., Fasoli M., Zadra C., Lovato A., Pezzotti M., Delledonne M., Polverari A. (2010): General and species-specific transcriptional responses to downy mildew infection in a susceptible (Vitis vinifera) and a resistant (V. riparia) grapevine species. BMC Genomics, 11: 117.

               Since KOF112 is not a pathogen in grapevine, actin remodelling might not be induced by KOF112 inoculation.

The information provided by the authors in this answer is not correct. In the provided references the authors cite to support their argument, the stability of ß-actine in grapevine during P. viticola-grapevine interaction was not assessed. In Polesani et al., 2010, ß-actin is only used as a reference gene for the qRT-PCR without confirming its appropriateness as control in their setting. There is no evidence in these articles that they tested the stability of ß-actine in P. viticola-grapevine interaction. In Figueiredo et al., 2012, they only referred to Polesani et al. 2010 with the following statement “Actin (TC81781) described as being stable in the P. viticola grapevine interaction (Polesani et al. 2010) and tested in our material (data not shown) was used as a reference gene for data normalization.”  Considering these old references (2010 and 2012), which are unclear regarding the proper testing of actin as reference, together with the known fact that actin is triggered upon MAMPs (which are present in the beneficial bacteria) and DAMPs perception (e.g., presented in 2020 in “New Phytologist”; https://onlinelibrary.wiley.com/doi/10.1111/nph.16584), I am not convinced that ß-actin is a suitable and stable reference gene in your study. I highly recommend to at least use another stable housekeeping gene as a reference for your qRT-PCR study to confirm the findings.

  1. My previous comment: The discussion is poor for many reasons:
  2. The authors should consult the latest reviews and research on biocontrol agents, cyclic lipopeptide, and plant-induced systemic resistance. To name a few, the group of Marc Ongena, Monica Hofta, and Corné Pieterse are experts in these fields for many years. However, none of their publications were consulted or cited here.
  3. The discussion is only a re-description of the results to justify the results obtained in this study
  4. results are not put in proper literature context
  5. there are no critical assessments of the data.

Authors’ answer: 1. We added deeper discussions using the following references, published in 2019-2021 years, in the revised manuscript.

Bruisson S, Zufferey M, L'Haridon F, Trutmann E, Anand A, Dutartre A, De Vrieze M, Weisskopf L. Endophytes and Epiphytes From the Grapevine Leaf Microbiome as Potential Biocontrol Agents Against Phytopathogens. Front Microbiol. 2019 Nov 29;10:2726. doi: 10.3389/fmicb.2019.02726. PMID: 31849878; PMCID: PMC6895011.

Li Y, Héloir MC, Zhang X, Geissler M, Trouvelot S, Jacquens L, Henkel M, Su X, Fang X, Wang Q, Adrian M. Surfactin and fengycin contribute to the protection of a Bacillus subtilis strain against grape downy mildew by both direct effect and defence stimulation. Mol Plant Pathol. 2019 Aug;20(8):1037-1050. doi: 10.1111/mpp.12809. Epub 2019 May 18. PMID: 31104350; PMCID: PMC6640177.

Leal C, Fontaine F, Aziz A, Egas C, Clément C, Trotel-Aziz P. Genome sequence analysis of the beneficial Bacillus subtilis PTA-271 isolated from a Vitis vinifera (cv. Chardonnay) rhizospheric soil: assets for sustainable biocontrol. Environ Microbiome. 2021 Jan 29;16(1):3. doi: 10.1186/s40793-021-00372-3. PMID: 33902737; PMCID: PMC8067347.

                  (See the revised Discussion section, please)

This comment is not well fulfilled. Adding three references to the discussion, one of which only added to a previously existing sentence from the former version of the manuscript, is not enrichment of the discussion. The discussion is poor and still only attempts to justify the results, with no proper critical assessment of the findings. There are many aspects that can be discussed to interpret the observations and the potential of your new strain to trigger plant defenses and having antifungal activities. To name a few, the capability of this bacterial strain to produce surfactin or bacillibactine. The results need to be put in the proper literature context to enrich the discussion and convince the readers with the potential of the new strain to antagonize major plant pathogens.

Moreover, ignoring publication/research by leading authors in the field is a questionable oversight. Here are just a few examples of other similar research in your field, which should be discussed:

- https://doi.org/10.1111/mpp.12809

- https://doi.org/10.3389/fmicb.2017.02552

- https://doi.org/10.3389/fpls.2020.594530

- https://doi.org/10.3390/su12239917

- https://doi.org/10.3389/fpls.2019.01112

- https://doi.org/10.1111/mpp.12170

- Heyman, Lisa, et al. “Potential of Pseudomonas Cyclic Lipopeptides to Control Downy Mildew in Grapevine by Induced Resistance and Direct Antagonism.” Molecular plant-microbe interactions, vol. 32, no. 10, Amer Phytopathological Soc, 2019, pp. 95–95.

Other comments:

On page 10 line 272*273, add a reference to support this statement.

Author Response

Point-by-point response to the comments (Reviewer #2):

Reviewers' comments:

Additional analysis has been done and substantial improvements have been made in this revision. Many of my previous questions/concerns have been addressed. There are however important issues that have not been properly addressed:

Answer: Thank you very much for your suggestions to improve our manuscript. The manuscript was revised according to your suggestions. The following is our point-by-point response to the comments and detailing all changes made on the revised manuscript. We hope that our answers to your comments are satisfactory and the revised manuscript is now acceptable for publication in the Plants.

  1. There are several studies about biocontrol agents in grapevine. Here are just two examples (Heyman, Lisa, et al. “Potential of Pseudomonas Cyclic Lipopeptides to Control Downy Mildew in Grapevine by Induced Resistance and Direct Antagonism.” MOLECULAR PLANT-MICROBE INTERACTIONS, vol. 32, no. 10, Amer Phytopathological Soc, 2019, pp. 95–95; “Surfactin and fengycin contribute to the protection of a Bacillus subtilis strain against grape downy mildew by both direct effect and defence stimulation” https://doi.org/10.1111/mpp.12809). It is crucial to discuss similar research in the field; otherwise, this gives an impression that the authors are intentionally omitting these studies to claim novelty. In addition, the background is too narrow by focusing only on describing the problem and naming the biofungal agents that are currently used and not talking about the known molecular background of these agents, such as the cyclic lipopeptides or antifungal compounds. Especially since the results highlighted the presence of the cyclic lipopeptides in the identified strain. It is crucial to provide background about these secondary metabolites that is a main feature of these beneficial bacteria (especially Bacillus velezensis). Cyclic lipopeptides are known to increase plant resistance and have antimicrobial activities, which is main part of your study.

Answer: We have taken your suggestions, and added a new paragraph describing relationship between cyclic lipopeptides and biological control in the revised Introduction section as follow:

‘Cyclic lipopeptides produced by biological control agents has received considerable attention as one of weapons for disease control in plants, since some cyclic lipopeptides also works as elicitors in plants as well as antimicrobial metabolites [16-18]. For example, fengycin and surfactin secreted by B. subtilis GLB191 contribute to protect grape downy mildew by direct effect against P. viticola and induction of plant defense responses [17]. Iturin A induces defense response in plants depending on its structure [19]. The cyclization of the seven amino acids and/or the β-hydroxy fatty acid chain of iturin A are required for the induction of defense response. Although no evidence of the mechanisms how cyclic lipopeptides trigger plant defense response in plants is available so far, microorganisms showing bifunctional activity against phytopathogens may be an innovative biological control agent in viticulture.’

(See p. 2, lines 72-82, please)

*References

  1. Heyman, L.; Ferrarini, E.; Omoboye, O.O.; Sanchez, L.; Barka, E.A.; Höfte, M. Potential of Pseudomonas cyclic lipopeptides to control downy mildew in grapevine by induced resistance and direct antagonism. In: MOLECULAR PLANT-MICROBE IN-TERACTIONS. American Phytopathological Society, St Paul, USA, 2019, p. 95.
  2. Li, Y.; Héloir, M.C.; Zhang, X.; Geissler, M.; Trouvelot, S.; Jacquens, L.; Henkel, M.; Su, X.; Fang, X.; Wang, Q.; Adrian, M. Surfactin and fengycin contribute to the protection of a Bacillus subtilis strain against grape downy mildew by both direct effect and defence stimulation. Mol. Plant Pathol. 2019, 20, 1037-1050.
  3. Omoboye, O.O.; Oni, F.E.; Batool, H.; Yimer, H.Z.; De Mot, R.; Höfte, M. Pseudomonas cyclic lipopeptides suppress the rice blast fungus Magnaporthe oryzae by induced resistance and direct antagonism. Front. Plant Sci. 2019, 10, 901.
  4. Kawagoe, Y.; Shiraishi, S.; Kondo, H.; Yamamoto, S.; Aoki, Y.; Suzuki, S. Cyclic peptide iturin A structure-dependently induces defense response in Arabidopsis plants by activating SA and JA signaling pathways. Biochem. Biophys. Res. Commun. 2015, 460, 1015-1020.

  1. This is not true. You can make boxplots with as few as three datapoints (although at least five should be used). Also, even on a par-chart graph you can easily add the datapoints. It is absolutely required to plot the datapoints since it shows where most of your observation lie on the range of the replicates. Otherwise, the graph oversimplifies the findings as it only shows the average, which is often skewed by outliers and non-normal distribution of the data. Therefore, this data representation can be misleading and omits crucial information to judge and interpret the data. Refer to e.g. 10.1371/journal.pbio.1002128 for clarification.

Answer: We agreed with you suggestion, and changed bar graphs to box plots in the revised Figures 5, 6, 7, and 8.

       (See the revised Figures, please)

  1. Your answer confirmed my concern that the phenotype that you see on the plant leaf discs is not due to increasing plant resistance but rather direct interaction/antagonism of the bacteria with the fungus. Particularly since you showed that the bacteria have antifungal activity in vitro and appear to harbor antifungal compounds against these fungi.

In addition, you use the same amount of bacteria (1 × 108 cfu/mL) in an unnatural setup (detached leaves) as is recommended for using the commercial biological fungicide in the field. I am concerned that the phenotype you are observing does not support the interpretation of increased plant resistance.

Answer: We have taken your suggestion, and added the following sentence in the revised Discussion section:

‘Since we didn’t analyze the biocontrol activity of KOF112 by field trials in vineyards, it is unclear whether the laboratory experiments using leaf disks does not support the interpretation of increased plant resistance. Answering the question by field trials in vineyards would make our hypothesis, that KOF112 also works as a biotic elicitor in pest management strategies, a stronger proposition.’

        (See p. 15, lines 347-352, please)

  1. The information provided by the authors in this answer is not correct. In the provided references the authors cite to support their argument, the stability of ß-actine in grapevine during P. viticola-grapevine interaction was not assessed. In Polesani et al., 2010, ß-actin is only used as a reference gene for the qRT-PCR without confirming its appropriateness as control in their setting. There is no evidence in these articles that they tested the stability of ß-actine in P. viticola-grapevine interaction. In Figueiredo et al., 2012, they only referred to Polesani et al. 2010 with the following statement “Actin (TC81781) described as being stable in the P. viticola grapevine interaction (Polesani et al. 2010) and tested in our material (data not shown) was used as a reference gene for data normalization.” Considering these old references (2010 and 2012), which are unclear regarding the proper testing of actin as reference, together with the known fact that actin is triggered upon MAMPs (which are present in the beneficial bacteria) and DAMPs perception (e.g., presented in 2020 in “New Phytologist”; https://onlinelibrary.wiley.com/doi/10.1111/nph.16584), I am not convinced that ß-actin is a suitable and stable reference gene in your study. I highly recommend to at least use another stable housekeeping gene as a reference for your qRT-PCR study to confirm the findings.

Answer: According to the following references, we performed some experiments to select a reference gen for real-time RT-PCR analysis. Since we used grapevine leaves (not grape berry) in our study, we selected ‘ubiquitin’ as a reference gene. Ubiquitin primers were in accordance with Bogs et al. 2005. The transcriptional profiles of genes encoding class IV chitinase and β-1,3-glucanase in grape leaves untreated or treated with KOF112 by real-time RT-PCR using ubiquitin were similar to those by real-time RT-PCR using actin as a reference gene in the former manuscript (See the following figure, please). However, as you pointed out, we changed former Figure 9 (actin was used as a reference gene) to the revised Figure 9 (ubiquitin was used as a reference gene).

Reid et al. 2006. An optimized grapevine RNA isolation procedure and statistical determination of reference genes for real-time RT-PCR during berry development. BMC Plant Biol. 6, 27.

Wei et al. 2021. Identification of optimal and novel reference genes for quantitative real-time polymerase chain reaction analysis in grapevine. Aust. J. Grape Wine Res. 27, 325-333.

Luo et al. 2018. Selection of reference genes for miRNA qRT-PCR under abiotic stress in grapevine. Sci. Rep. 8, 4444.

Bogs et al. 2005. Proanthocyanidin synthesis and expression of genes encoding leucoanthocyanidin reductase and anthocyanidin reductase in developing grape berries and grapevine leaves. Plant Physiol. 139, 652-663.

Maia et al. 2020. Integrating metabolomics and targeted gene expression to uncover potential biomarkers of fungal/oomycetes-associated disease susceptibility in grapevine. Sci. Rep. 10, 15688.

  1. This comment is not well fulfilled. Adding three references to the discussion, one of which only added to a previously existing sentence from the former version of the manuscript, is not enrichment of the discussion. The discussion is poor and still only attempts to justify the results, with no proper critical assessment of the findings. There are many aspects that can be discussed to interpret the observations and the potential of your new strain to trigger plant defenses and having antifungal activities. To name a few, the capability of this bacterial strain to produce surfactin or bacillibactine. The results need to be put in the proper literature context to enrich the discussion and convince the readers with the potential of the new strain to antagonize major plant pathogens.

Moreover, ignoring publication/research by leading authors in the field is a questionable oversight. Here are just a few examples of other similar research in your field, which should be discussed:

- https://doi.org/10.1111/mpp.12809

- https://doi.org/10.3389/fmicb.2017.02552

- https://doi.org/10.3389/fpls.2020.594530

- https://doi.org/10.3390/su12239917

- https://doi.org/10.3389/fpls.2019.01112

- https://doi.org/10.1111/mpp.12170

- Heyman, Lisa, et al. “Potential of Pseudomonas Cyclic Lipopeptides to Control Downy Mildew in Grapevine by Induced Resistance and Direct Antagonism.” Molecular plant-microbe interactions, vol. 32, no. 10, Amer Phytopathological Soc, 2019, pp. 95–95.

Answer: We have taken your suggestions, and revised the Discussion section all over by adding proper references.

        (See the revised Discussion section, please)

  1. On page 10 line 272*273, add a reference to support this statement.

Answer: We added the following reference for the statement.

        ‘30. Borriss, R.; Use of plant-associated Bacillus strains as biofertilizers and biocontrol agents. In Bacteria in Agrobiology: Plant Growth Responses, Maheshwari, D.K. Eds.; Springer, Berlin, Heidelberg, 2011, pp. 41-76.’

        (See p. 15, line 310, please)

Reviewer 3 Report

In this version of the manuscript, authors have addressed a number of concerns raised by this reviewer, as the incorporation of control images of untreated fungi and of a Figure showing the comparative analysis of BGCs in related B. velezensis strains. Also, several related articles have been included in the Discussion. These changes are fine.

However, authors have decided not to carry out important suggested control experiments in planta using an isolate or strain that does not inhibit the pathogenic fungi in vitro This was an important control,as judged by this reviewer, to provide support to different claims of the authors with regards to the biocontrol ability of strain KOF112.

In response to this suggestion of including a "disarmed" bacterial strain, the authors argue that including a saprophytic bacterium would ward off the paper from the objective of the study.

Nevertheless, as textually mentioned in the article, "The objective of this study was to clarify the possibility of using endophytic bacteria as biological control agents in biofungicides used in viticulture".

So, considering that the authors claim that the observed protection in plant experiments is due to the production of secondary metabolites with antifungal activity, on the one hand, and also due to induction of a systemic resistance response, on the other hand, I consider that these claims need to be supported by either the use of mutants derived from strain KOF112 that lack antifungal activity or do not induce ISR, or, alternatively by comparing the performance of bacterial isolates or strains with no antifungal activity. A simple solution would be to use a non-pathogenic and lab strain like E. coli DH5alfa or similar, which lacks BGCs for secondary metabolites, and do not have a plant colonizing habit. Perhaps the better option would be to use one of the endophytic isolates from grapevine tissues, that dit not show antifungal activity in the initial screening that resulted in KOF112. The control isolate/strain is important for contrasting the specificity of the effects shown in Figures 1, 5, 6, 7, 8 and 9. Otherwise, I can hardly see a reason for using strain KOF112 and any other bacterium.

Finally, another relevant issue was that in the original article authors stated that they "...cannot demonstrate any positive results of KOF112 colonization in grapevine". Now, instead of claryfing this statement, as requested by this reviewer, the authors have decided to eliminate that sentence. However, my suggestion was not to remove the sentence, but to explain it. Such explanation is still lacking. Colonization of grapevine tissues by this strain would be a relevant trait to promote its use in a commercial product.

Author Response

Point-by-point response to the comments (Reviewer #3):

Reviewers' comments:

  1. In this version of the manuscript, authors have addressed a number of concerns raised by this reviewer, as the incorporation of control images of untreated fungi and of a Figure showing the comparative analysis of BGCs in related B. velezensis strains. Also, several related articles have been included in the Discussion. These changes are fine.

Answer: Thank you very much for your suggestions to improve our manuscript. The manuscript was revised according to your suggestions. The following is our point-by-point response to the comments and detailing all changes made on the revised manuscript. We hope that our answers to your comments are satisfactory and the revised manuscript is now acceptable for publication in the Plants.

  1. However, authors have decided not to carry out important suggested control experiments in planta using an isolate or strain that does not inhibit the pathogenic fungi in vitro This was an important control,as judged by this reviewer, to provide support to different claims of the authors with regards to the biocontrol ability of strain KOF112.

In response to this suggestion of including a "disarmed" bacterial strain, the authors argue that including a saprophytic bacterium would ward off the paper from the objective of the study.

Nevertheless, as textually mentioned in the article, "The objective of this study was to clarify the possibility of using endophytic bacteria as biological control agents in biofungicides used in viticulture".

So, considering that the authors claim that the observed protection in plant experiments is due to the production of secondary metabolites with antifungal activity, on the one hand, and also due to induction of a systemic resistance response, on the other hand, I consider that these claims need to be supported by either the use of mutants derived from strain KOF112 that lack antifungal activity or do not induce ISR, or, alternatively by comparing the performance of bacterial isolates or strains with no antifungal activity. A simple solution would be to use a non-pathogenic and lab strain like E. coli DH5alfa or similar, which lacks BGCs for secondary metabolites, and do not have a plant colonizing habit. Perhaps the better option would be to use one of the endophytic isolates from grapevine tissues, that dit not show antifungal activity in the initial screening that resulted in KOF112. The control isolate/strain is important for contrasting the specificity of the effects shown in Figures 1, 5, 6, 7, 8 and 9. Otherwise, I can hardly see a reason for using strain KOF112 and any other bacterium.

Answer: We have taken your suggestion, and included additional data with an endophytic isolate, isolated from grapevine, with no antifungal activity in the revised Figures 1, 5, 6, 8 and 9. Agrobacterium sp. isolate CHB3, selected as a control isolate with no antifungal activity, exhibited no suppressive effect on mycelial growth of all the fungi tested, no inhibitory effect on all the plant disease tested and no effect on zoospore release from zoosporangia and zoospore germination of P. viticola (the revised Figures 1, 5, 6, 7, and 8). Also, Agrobacterium sp. isolate CHB3 didn’t upregulate the transcription of both genes encoding class IV chitinase and β-1,3-glucanase in grape leaves (the revised Figure 9). As pointed out by you, these data support our s claims with regards to the biocontrol ability of strain KOF112.

         (See the revised Figures and Results section, please)

  1. Finally, another relevant issue was that in the original article authors stated that they "...cannot demonstrate any positive results of KOF112 colonization in grapevine". Now, instead of claryfing this statement, as requested by this reviewer, the authors have decided to eliminate that sentence. However, my suggestion was not to remove the sentence, but to explain it. Such explanation is still lacking. Colonization of grapevine tissues by this strain would be a relevant trait to promote its use in a commercial product.

Answer: We have taken your suggestion, and added the explanation about the sentence as follow:

        ‘Although KOF112 can induce plant defense responses in grapevine, we cannot demonstrate any positive results of KOF112 colonization in grapevine so far. Therefore, further investigation employing scanning electron microscopic analysis would reveal whether KOF112 colonizes on foliar-sprayed leaves.’

        (See p. 16, lines 370-373, please) 

Round 3

Reviewer 3 Report

In this version of the article, the authors have now included important control treatments in the evaluation of the impact of KOF112 inoculation on different in vitro or in planta tests. Now, it is clear that the effects can be ascribed specifically to the studied isolate KOF112, and that they are not just due to the presence of any bacterium at a high cell density. I consider that the article is now acceptable in the current version. I just recommend a thorough revision of english style to polish the text.

Author Response

Point-by-point response to the comments (Reviewer #3):

Reviewers' comments:

In this version of the article, the authors have now included important control treatments in the evaluation of the impact of KOF112 inoculation on different in vitro or in planta tests. Now, it is clear that the effects can be ascribed specifically to the studied isolate KOF112, and that they are not just due to the presence of any bacterium at a high cell density. I consider that the article is now acceptable in the current version. I just recommend a thorough revision of english style to polish the text.

Answer: Thank you very much for your advice to improve our manuscript. In the revised manuscript, the editing of English language and style to polish our manuscript was performed by native speakers.

We hope that our revision is satisfactory and the revised manuscript is now acceptable for publication in the Plants.
